# Conformal Information Pursuit for Interactively Guiding Large Language Models

**Kwan Ho Ryan Chan**[*]   **Yuyan Ge**   **Edgar Dobriban**   **Hamed Hassani**   **René Vidal**

University of Pennsylvania

## Abstract

A significant use case of instruction-finetuned Large Language Models (LLMs) is to solve question-answering tasks interactively. In this setting, an LLM agent is tasked with making a prediction by sequentially querying relevant information from the user, as opposed to a single-turn conversation. This paper explores sequential querying strategies that aim to minimize the expected number of queries. One such strategy is Information Pursuit (IP), a greedy algorithm that at each iteration selects the query that maximizes information gain or equivalently minimizes uncertainty. However, obtaining accurate estimates of mutual information or conditional entropy for LLMs is very difficult in practice due to over- or under-confident LLM probabilities, which leads to suboptimal query selection and predictive performance. To better estimate the uncertainty at each iteration, we propose *Conformal Information Pursuit (C-IP)*, an alternative approach to sequential information gain based on conformal prediction sets. More specifically, C-IP leverages a relationship between prediction sets and conditional entropy at each iteration to estimate uncertainty based on the average size of conformal prediction sets. In contrast to conditional entropy, we find that conformal prediction sets are a distribution-free and robust method of measuring uncertainty. Experiments with 20 Questions show that C-IP obtains better predictive performance and shorter query-answer chains compared to previous approaches to IP and uncertainty-based chain-of-thought methods. Furthermore, extending to an interactive medical setting between a doctor and a patient on the MediQ dataset, C-IP achieves competitive performance with direct single-turn prediction while offering greater interpretability.

## 1   Introduction

Large Language Models (LLMs) [2, 3, 7, 42, 83] fine-tuned on instruction datasets are capable of generating human-like dialogues. Although many use-cases constrain these models to single-turn interactions [99, 111, 138], where a single prompt typically provides all context and examples, more realistic conversations behave interactively [12, 70]. In this work, we are interested in the setting where LLMs interactively solve a prediction task in a multi-turn, information-seeking environment by asking queries and obtaining information sequentially [71]. Popular examples of such settings include the game of 20 Questions (20Q) [11, 84, 143] and Interactive Medical Question Answering [70] (Figure 1).

Figure 1: Example of diagnosis via Patient and Doctor LLM interaction.

---

[*]Corresponding author. Email: `ryanckh@seas.upenn.edu`

39th Conference on Neural Information Processing Systems (NeurIPS 2025).

While there are many strategies to interactively arrive at a final prediction, we consider strategies that yield a small number of queries. One such strategy is to always ask the most "informative" query given the prior history, until the model is confident enough to make a prediction. This process is characterized in a framework called Information Pursuit (IP) [16, 38], which defines the most informative query as the one that *maximizes information gain* at each iteration. This requires selecting the query whose answer has the maximum mutual information [18, 21, 30, 77, 145] with the prediction, given the history of prior query-answer pairs. Equivalently, one can select the query whose answer minimizes the uncertainty of the prediction, as quantified by the conditional entropy.

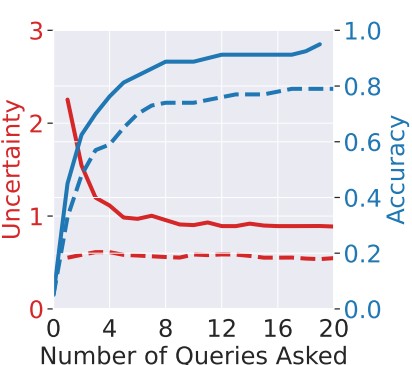

Figure 2: IP with calibrated (solid) versus uncalibrated (dashed) measures of uncertainty.

In practice, implementing IP for LLMs requires estimating the conditional entropy, which is very challenging because the probabilities of LLM outputs, such as logits of output tokens, are often *miscalibrated*: LLM predictions are often strongly over- or under-confident [6, 31, 39, 55, 78, 109, 113, 118, 121, 128, 132, 142, 144, 147, 148]. Consequently, using probability estimates to measure uncertainty leads to inaccurate estimates of information and lower predictive performance. We illustrate this phenomenon in Figure 2 using the example of two LLMs playing 20 Questions. In this setting, the Querier LLM has to guess the object that the Answerer LLM is thinking of by asking a sequence of most informative questions. Ideally, the uncertainty and accuracy of the prediction should behave like the solid red and solid blue curves. At the first iteration, since the Querier LLM has not gathered sufficient information, the uncertainty of the prediction should be high. As iterations proceed, uncertainty should diminish and converge to a certain level, where any new query would no longer offer any additional information than what the Querier LLM already knows. Then, the Querier LLM should stop and make a prediction. However, we observe that using the LLM's output probabilities to estimate uncertainty yields a flat and uninformative curve (dashed red curve). Naturally, this leads to suboptimal query selection and lower predictive performance (dashed blue curve). This motivates the need for better uncertainty estimation.

To obtain more accurate measures of uncertainty for the outputs of the LLM, we propose an alternative approach to sequential information gain that leverages the notion of *prediction sets* from predictive inference and conformal prediction [5, 124]. Although the history of prediction sets dates back to the 1940s [103, 119, 120, 125, 133], the use of the expected size of a prediction set as a measure of uncertainty has been studied and popularized only recently [4, 28, 32, 58, 82, 100, 101]. In more recent applications, it has also been used to calibrate the LLMs' output probabilities for a more accurate measure of model confidence [23, 61, 80, 95, 140]. By and large, standard entropy as a measure of uncertainty relies heavily on proper estimation of the target distribution, while prediction sets from conformal prediction allow for a distribution-free and instance-wise measure, offering more flexibility, robustness, and interpretability. Given its dual purpose of being a tool for uncertainty quantification and calibration, we are motivated to use prediction sets as a measure of informativeness in lieu of conditional entropy.

Therefore, in this paper we propose **Conformal Information Pursuit (C-IP)**, a greedy sequential approach that asks informative queries in the order of information gain by minimizing prediction set size instead of the standard entropy. We first leverage the relationship between entropy and the expected size of prediction sets using results from Correia et al. [27], allowing us to upper bound the entropy of the target distribution given the history of query-answer pairs by the expected logarithm of the size of the prediction set for the ground-truth answer given the history. Then, utilizing conformal prediction, we propose two ways of constructing sequences of prediction sets, which allows us to provide accurate measures of informativeness for any given query at any iteration of the algorithm. To demonstrate the properties and behavior of our method, we perform a thorough study of our method in the setting of 20Q. Finally, we apply our method to the setting of interactive medical question answering on the MediQ [70] dataset. Our results demonstrate not only the applicability of our method in real-world settings, but also its competitive predictive performance. Our code is available at https://www.github.com/ryanchankh/ConformalInformationPursuit/.

## 2 Information Pursuit: A Framework for Sequential Information Gain

We first review the Information Pursuit (IP) framework [16, 38, 51], which will help us lay the groundwork for developing a greedy algorithm that asks queries using the information gain criterion [21, 30, 51, 145]. Throughout the paper, $(X, Y)$ denotes an input-output pair, where $X : \Omega \to \mathcal{X}$ and $Y : \Omega \to \mathcal{Y}$ are random variables, $\Omega$ is the sample space, and $\mathcal{X}$ and $\mathcal{Y}$ are the sets of values that they can take on. We denote by $I(X; Y)$ the mutual information between $X$ and $Y$, by $H(X)$ the entropy of the random variable $X$, and by $H(Y \mid X)$ the conditional entropy of $X$ given $Y$.

**Setup.** The modern generative version of IP is a framework for making interpretable predictions: The user defines a set $\mathcal{Q}$ of task-relevant queries. Each query $q \in \mathcal{Q}$ is a function $q : \mathcal{X} \to \mathcal{A}$ that maps the textual input $x \in \mathcal{X}$ to an answer $q(x) \in \mathcal{A}$. For example, for the task of disease classification, $\mathcal{X}$ contains the set of patient features (e.g., clinical notes), $\mathcal{Y}$ is the set of disease labels (e.g., "pneumonia", "cardiomegaly"), $\mathcal{Q}$ is a set of queries about the patient such as "Are you coughing?" or "Do you have fever?", and $\mathcal{A}$ to be the set of possible query answers (e.g. "Yes, I do", "The patient feels warm."). Last but not least, for our question-answering task, all query-answers $\mathcal{A}$ and labels $\mathcal{Y}$ are a subset of the space of natural language $\mathcal{X}$.

An important desired property of the query set is *sufficiency*, which states that for all $x, y$,

$$\mathbb{P}(y \mid x) = \mathbb{P}(y \mid \mathcal{Q}(x)) = \mathbb{P}(y \mid \{x' \in \mathcal{X} : q(x) = q(x'), \ \forall q \in \mathcal{Q}\}). \tag{1}$$

That is, the answers to all queries about $x$, $\mathcal{Q}(x) := \{q(x) : q \in \mathcal{Q}\}$, determine the *posterior* $\mathbb{P}(y \mid x)$. In this work, we consider obtaining $\mathcal{Q}$ either from an offline set of answerable questions based on the task, or by prompting an LLM pre-trained with open-domain knowledge, hence capable of generating any necessary query. In either case, $\mathcal{Q}$ is expected to satisfy sufficiency.

**IP Algorithm.** Given an input $x^{\mathrm{obs}} \in \mathcal{X}$, the IP algorithm makes a prediction for $Y$ by sequentially selecting queries $q_1, q_2, \dots \in \mathcal{Q}$ in order of *information gain*, i.e., by maximizing the conditional information between a query answer and the prediction given the *history* of prior query-answer pairs:

$$q_1 = \mathrm{IP}(\emptyset) = \arg\max_{q \in \mathcal{Q}} I(q(X); Y);$$
$$q_{k+1} = \mathrm{IP}(\{q_i, q_i(x^{\mathrm{obs}})\}_{1:k}) = \arg\max_{q \in \mathcal{Q}} I(q(X); Y \mid q_{1:k}(x^{\mathrm{obs}})). \tag{IP}$$

When $k = 0$, the history of prior query-answer pairs is empty, i.e., $q_{1:k} := \emptyset$, and IP selects the query whose answer is most informative for predicting $Y$. When $k \geq 1$, IP selects the most informative query conditioned on the *history* $q_{1:k}(x^{\mathrm{obs}})$[2]. IP terminates at iteration $L$ if $\forall q_{k+1} \in \mathcal{Q}, I(q_{k+1}(X); Y \mid q_{1:L}(x^{\mathrm{obs}})) = 0$, meaning that no additional query provides extra information about the target $Y$. In practice, evaluating $\mathbb{P}(Y \mid q_{1:k}(x^{\mathrm{obs}}))$ at test time allows us to observe how each query and answer affects the prediction at each iteration $k = 1, \dots, L$.

**IP for LLMs.** Previous work of IP [16, 17, 19, 37, 60] involves learning the joint distribution $\mathbb{P}(Y, \mathcal{Q}(X))$ (or the conditional distribution $\mathbb{P}(Y \mid q_{1:k}(x))$) from training data such that samples of $\mathbb{P}(Y \mid q_{1:k}(x))$ can be obtained efficiently. In contrast, here we assume that the conditional distribution is given by an LLM directly: We construct a probabilistic predictor[3] $f : \mathcal{X} \to \Delta(\mathcal{Y})$, which takes a text prompt as input and outputs a set of probability scores for each class. In particular, we first extract the logits of the output tokens of interest (such as class names), and then apply a softmax transform to obtain a predicted distribution (More details in Appendix §G.5).

Since $\mathbb{P}$ is given in this case, we can simplify by writing the equivalent formulation of (IP) as conditional entropy minimization, for each $k$:

$$\operatorname*{argmax}_{q \in Q} I(q(X); Y \mid q_{1:k}(x^{\mathrm{obs}})) = \operatorname*{argmax}_{q \in Q} \{H(Y \mid q_{1:k}(x^{\mathrm{obs}})) - H(Y \mid q(X), q_{1:k}(x^{\mathrm{obs}}))\}$$
$$= \operatorname*{argmin}_{q \in Q} H(Y \mid q(X), q_{1:k}(x^{\mathrm{obs}})). \tag{2}$$

In other words, at any iteration, the query that maximizes the conditional mutual information also minimizes the conditional entropy. To find the query in (2), one would have to collect samples, estimate $H(Y \mid q_{1:k}(x^{\mathrm{obs}}))$ for each $q$, and select the query that minimizes the conditional entropy.

---

[2]Note that $\mathbb{P}(\cdot \mid q_{1:k}(x^{\mathrm{obs}})) = \mathbb{P}(\cdot \mid \{x' \in \mathcal{X} : q_i(x') = q_i(x^{\mathrm{obs}}) \text{ for all } i = 1, \dots, k\})$

[3]Here $\Delta(\mathcal{Y})$ is the set of probability distributions over $\mathcal{Y}$.

However, as we have argued in §1, since $f$ is not trained to model $\mathbb{P}(Y, \mathcal{Q}(X))$ but is extracted from an LLM that models $P_{\mathrm{LLM}}(X)$ as a general language distribution, the output probabilities are often miscalibrated, motivating our need for an alternative approach to IP.

# 3 Conformal Information Pursuit: Information Gain with LLMs

## 3.1 Uncertainty Estimation via Predictive Inference

Next, we propose an alternative approach which leverages recent developments in predictive inference and conformal prediction [5, 124]. Specifically, the results of Correia et al. [27] allow us to upper bound entropy by the size of certain prediction sets for the ground-truth answers, hence measuring uncertainty in a distribution-free and robust manner. We now turn to introducing this approach.

**Preliminary on Predictive Inference.** Consider a function $f$ that assigns to each value $x$ the predicted probabilities $f(x) = (f(x)_1, \ldots, f(x)_{|\mathcal{Y}|})$ for the classes $1, \ldots, |\mathcal{Y}|$. We may assume that $f$ is pre-trained with data drawn from some unknown-distribution $P_{\mathrm{data}}$. Predictive inference, generally speaking, seeks to provide guarantees about the prediction $f(X_{\mathrm{test}})$ of a new test sample $(X_{\mathrm{test}}, Y_{\mathrm{test}}) \sim P_{\mathrm{data}}$. One way of doing so is using *prediction sets*: For a threshold $\tau \in [0, 1]$, a prediction set function $\mathcal{C}_\tau : \mathcal{X} \to 2^{|\mathcal{Y}|}$ selects labels with predicted probabilities $f(x)_y$ larger than $\tau$:

$$\mathcal{C}_\tau(x) = \{y \in \mathcal{Y} \mid f(x)_y > \tau\}^4. \tag{3}$$

Then, we seek to guarantee that the true label $Y_{\mathrm{test}}$ is in the prediction set $\mathcal{C}_\tau$ with probability $1 - \alpha$.

While there are a number of ways to set the threshold $\tau$, conformal prediction [102, 123] was introduced as a non-parametric, finite-sample way to achieve such guarantee. Specifically, we will introduce the method of Split Conformal Prediction (SCP) [88].

Let $\mathcal{D}_{\mathrm{cal}} = \{(X_i, Y_i), i \in [N]\}$, where $(X_i, Y_i) \sim P_{\mathrm{data}}$ and $[N] := \{1, \ldots, N\}$, be a calibration dataset of feature-label, independent of the training samples of $f$. By only assuming the exchange-ability of the values $f(X_i)_{Y_i}, i \in [N] \cup \{\mathrm{test}\}$, SCP satisfies the so-called *marginal guarantee*:

$$\mathbb{P}_{X_{\mathrm{test}}, Y_{\mathrm{test}}, \mathcal{D}_{\mathrm{cal}}} \left(Y_{\mathrm{test}} \in \mathcal{C}_{\hat{\tau}}(X_{\mathrm{test}})\right) \geq 1 - \alpha, \tag{4}$$

by setting $\hat{\tau}$ as the $\lceil (1 - \alpha)(N + 1) \rceil$-th smallest of the values $f(X_i)_{Y_i}, i \in [N]^5$. Even if the distribution of $f(X)_Y$ when $(X, Y) \sim P_{\mathrm{data}}$ does not have any point masses, then the coverage probability is also upper bounded by $1 - \alpha_N$ for $\alpha_N = \alpha - 1/(N + 1)$. Hence, $\mathcal{C}_\tau(x)$ here is also commonly referred to a *conformal prediction set*. As we will see later, while SCP assumptions are fairly mild, obtaining a calibration set from the same data distribution $P$ in the sequential setting poses additional challenges.

**Contrasting Notions of Uncertainty.** There is a plethora of work that uses notions from predictive inference, such as the *size* of a prediction set, as a means to measure uncertainty [4, 28, 32, 58, 82, 100, 101]. We seek to mathematically characterize the relation between entropy and the size of prediction sets by leveraging a connection provided by Correia et al. [27] via the following proposition:

**Proposition 3.1** (Correia et al. [27], simplified)**.** For $\alpha \in (0, 0.5)$, consider any prediction set function $\mathcal{C}_{\hat{\tau}}$ satisfying

$$1 - \alpha \leq \mathbb{P}_{X, Y, \mathcal{D}_{\mathrm{cal}}} \left(Y \in \mathcal{C}_{\hat{\tau}}(X)\right) \leq 1 - \alpha_N. \tag{5}$$

Let $\lambda_\alpha := h_b(\alpha) + \alpha \log|\mathcal{Y}| - (1 - \alpha_N) \log(1 - \alpha)$, where $h_b$ is the binary entropy function. For the true distribution $P_{\mathrm{data}}$, we have

$$H(Y \mid X) \leq \lambda_\alpha + (1 - \alpha_N) \log \mathbb{E}_X[|\mathcal{C}_{\hat{\tau}}(X)|]. \tag{6}$$

In words, the uncertainty of $Y$ given data $X$ is upper bounded by the expected size of the prediction set. The more uncertain one is about $Y$, the higher the entropy and the larger the prediction set size, and vice versa. The proposition also tells us the following: whatever the construction of the prediction set may be, as long as it satisfies the marginal guarantee, the relationship between entropy and prediction set holds. Next, using this fact, we will derive a *conformalized* formulation for IP.

---

[4] The definition typically requires defining a nonconformity score $\psi : \mathcal{X} \times \mathcal{Y} \to \mathbb{R}$. Here we directly define the $\psi(x, y) = -P(y \mid x)$ for simplicity.

[5] Which can be obtained via sample quantiles. See Algorithm 1.

## 3.2 Conformal Information Pursuit Algorithm

We will now formalize our proposed method. Since we leverage SCP to obtain prediction sets at test time, we refer to our new approach to IP as **Conformal Information Pursuit (C-IP)**[6]. Throughout our work, we assume our prediction set takes the form

$$\mathcal{C}_\tau\Big(q_{1:k}(x)\Big) = \Big\{y \in \mathcal{Y} \mid \hat{P}\Big(Y \mid q_{1:k}(x)\Big) := f(q_{1:k}(x))_y > \tau\Big\}. \tag{7}$$

Since both the query answers $q(x)$ and the query-answer chains $q_{1:k}(x)$ are represented as text, they both live in the domain of $\mathcal{X}$. In the following, we will use the results in §3.1 to derive a new method for choosing the next query given history. For clarity, we use $\tau(\cdot)$ to denote $\tau$'s dependencies.

**C-IP: First Query $q_1$.** To select the first query, recall from (2) that $\mathrm{argmax}_{q \in \mathcal{Q}} I(q(X); Y) = \mathrm{argmin}_{q \in \mathcal{Q}} H(Y \mid q(X))$. Therefore, we can use Proposition 3.1 to construct a $\mathcal{C}_{\hat{\tau}(q)}$ for every $q \in \mathcal{Q}$ that satisfies the marginal coverage guarantee (5):

$$1 - \alpha \leq \mathbb{P}_{X,Y,\mathcal{D}_{\mathrm{cal}}}\big(Y \in \mathcal{C}_{\hat{\tau}(q)}(q(X))\big) \leq 1 - \alpha_N. \tag{8}$$

Then,

$$\min_{q \in \mathcal{Q}} H(Y \mid q(X)) \leq \min_{q \in \mathcal{Q}} \log \mathbb{E}_X\Big[|\mathcal{C}_{\hat{\tau}(q)}\big(q(X)\big)|\Big]. \tag{9}$$

In words, C-IP selects the first query $q_1$ by minimizing the above upper bound on the entropy.

**C-IP: Subsequent Queries $q_k$.** Similarly, for the subsequent iterations $k \geq 1$, due to (2), $\mathrm{argmax}_{q \in \mathcal{Q}} I(q(X); Y \mid q_{1:k}(x^{\mathrm{obs}})) = \mathrm{argmin}_{q \in \mathcal{Q}} H(Y \mid q(X), q_{1:k}(x^{\mathrm{obs}}))$. Again, from Proposition 3.1, we construct a $\mathcal{C}_{\hat{\tau}(q,q_{1:k}(x^{\mathrm{obs}}))}$ for every $q \in \mathcal{Q}$ that satisfies the coverage guarantee (5) conditioned on *the obtained* $q_{1:k}(x^{\mathrm{obs}})$, i.e.,

$$1 - \alpha \leq \mathbb{P}_{X,Y,\mathcal{D}_{\mathrm{cal}}}\big(Y \in \mathcal{C}_{\hat{\tau}(q,q_{1:k}(x^{\mathrm{obs}}))}(q(X)) \mid q_{1:k}(x^{\mathrm{obs}})\big) \leq 1 - \alpha_N. \tag{10}$$

Then,

$$\min_{q \in \mathcal{Q}} H(Y \mid q(X), q_{1:k}(x^{\mathrm{obs}})) \leq \min_{q \in \mathcal{Q}} \log \mathbb{E}_X\Big[|\mathcal{C}_{\hat{\tau}(q,q_{1:k}(x^{\mathrm{obs}}))}(q(X))| \mid q_{1:k}(x^{\mathrm{obs}})\Big]. \tag{11}$$

Our C-IP method aims to choose $q_k$ by minimizing the above upper bound. We will refer to the algorithm that uses empirical estimates of the entropy on the left-hand side of (11) as IP and to the algorithm that uses empirical estimates of the upper bound on the right-hand side of (11) as C-IP.

From our derivation above, directly applying Proposition 3.1 requires coverage guarantees in (8) and (10) that condition on the *one instantiation of the history* $q_{1:k}(x^{\mathrm{obs}})$ obtained from running C-IP. Unfortunately, this is intractable in general [59, 64], as the number of possible $q_{1:k}(x^{\mathrm{obs}})$ is exponentially large, of size $\sim |\mathcal{Q}|^k$. Even when the maximum number of iterations $L$ is small, ensuring this would also in general require a large enough subset of the calibration dataset $X_i$, $i \in [N]$, with the same history $q_{1:k}(x^{\mathrm{obs}})$. Therefore, we propose a related, but relaxed, formulation.

**Obtaining Prediction Sets $\mathcal{C}$.** To ensure tractability, we propose the following guarantee:

$$1 - \alpha \leq \mathbb{P}_{X,Y,Q_{1:k},\mathcal{D}_{\mathrm{cal}}}\Big(Y \in \mathcal{C}_{\hat{\tau}(k)}(Q_{1:k}(X))\Big) \leq 1 - \alpha_N, \quad \text{for } k = 1, \ldots, L, \tag{12}$$

which means to offer a guarantee over the distribution of random query-answer chains $Q_{1:k}(X)$ *with length $k$* rather than a single choice of $q_{1:k}(x^{\mathrm{obs}})$. As a result, our marginalized guarantee only requires us to construct $L$ prediction set functions. We refer this as *length marginalization*.

With length marginalization in mind, we now turn the discussion to obtain samples from $Q_{1:k}$ to construct $\mathcal{C}_{\hat{\tau}(k)}$ and satisfy (12). The question of how to efficiently sample histories arises in many occasions when implementing IP (see Related Work in Appendix §B for discussion). In this work, we consider two possible parameterizations, adaptive to the possible choices of the query set $\mathcal{Q}$:

---

[6] We remark that "conformalized" often refers to a wrapper of a method that enjoys a conformal guarantee; this is not the sense in which we use it.

- **Sampling Histories from a Uniform Parameterization:** When the query set $\mathcal{Q}$ is a closed set with pre-determined queries, an effective strategy is to parameterize queries at each iteration with a uniform distribution over the query set. As for V-IP [17], we propose to marginalize our coverage over *uniformly sampled histories of length $k$*. At each iteration $k = 1, \ldots, L$, where $L$ is the maximum number of iterations allowed, let $Q_{1:k} = (Q_1, \cdots, Q_k)$ be a selection of $k$ queries sampled uniformly at random from the set $\mathcal{Q}$, i.e.

$$Q_{1:k} \sim \text{Unif}(\mathcal{Q})^k \quad \text{for} \quad k = 1, \ldots, L. \tag{13}$$

- **Sampling Histories from LLM Simulations:** If $\mathcal{Q}$ is an open query set with the space of questions in natural language, another option is to consider the distribution of histories obtained from LLMs. For a given observation $x^{\text{obs}}$ and query chain $q_{1:k}$, $q_{k+1}$ can be obtained by directly prompting for the next query, i.e., $q_{k+1} \sim P_{\text{LLM}}(\mathcal{Q}, q_{1:k}(x^{\text{obs}}))$. We refer this strategy as Direct Prompting (DP) (see Appendix §I for the algorithm), where the distribution of queries is:

$$Q_{1:k} \sim \text{DP}(X, Y) \quad \text{for} \quad k = 1, \ldots, L. \tag{14}$$

Hence, we obtain a list of prediction set functions $\mathcal{C}_{\hat{\tau}(1)} \ldots \mathcal{C}_{\hat{\tau}(L)}$ for every length. The precise procedure is described in Appendix §I, Algorithm 1.

**C-IP Algorithm.** To summarize, the C-IP algorithm is as follows. First, we obtain a list of prediction set functions $\mathcal{C}_{\hat{\tau}_1} \ldots \mathcal{C}_{\hat{\tau}_L}$ based on the desired coverage $1 - \alpha$. Then, for any given observation $x^{\text{obs}}$,

$$q_1 = \text{C-IP}(\emptyset) = \underset{q \in \mathcal{Q}}{\arg\min} \log \mathbb{E}_X[|\mathcal{C}_{\hat{\tau}(1)}(q(X))|];$$
$$q_{k+1} = \text{C-IP}(\{q_i, q_i(x^{\text{obs}})\}_{1:k}) = \underset{q \in \mathcal{Q}}{\arg\min} \log \mathbb{E}_X[|\mathcal{C}_{\hat{\tau}(k+1)}(q(X))|\ q_{1:k}(x^{\text{obs}})]. \tag{C-IP}$$

Similar to IP, the quantities in expectation are estimated by their empirical expectations. See Appendix §I for pseudocode for the IP, C-IP and DP algorithms.

*Remark 1: Stopping Criteria.* As mentioned in §2, IP stops when $\forall q \in \mathcal{Q}, I(Y; q(X) \mid q_{1:k}(x^{\text{obs}})) = 0$. From (2), this corresponds to $\forall q_{k+1} \in \mathcal{Q}, H(Y \mid q_{1:k+1}(x^{\text{obs}})) - H(Y \mid q_{1:k}(x^{\text{obs}})) = 0$ (also referred to as the stability criterion [17]. This also nicely extends to C-IP: When the model is highly confident, the prediction set should be on average a *singleton*, which implies $\forall q_{k+1} \in \mathcal{Q}, \log \mathbb{E}_X[|\mathcal{C}_{\hat{\tau}(k+1)} q_{1:k+1}(k+1)|] - \log \mathbb{E}_X[|\mathcal{C}_{\hat{\tau}(k)}(q_{1:k}(X))|] = 0$. In other words, the stopping criteria are equivalent in IP and C-IP. In practice, this stringent stopping criterion may not be reached. Hence, we stop querying once the standard deviation of the estimated entropy for every query is smaller than a chosen threshold $\varepsilon > 0$. See Appendix §G.10 for more details.

*Remark 2: Risk Control.* Our marginalization approach is a means to estimate uncertainty. Ideally, one would characterize the full distribution over IP histories. We present our work as a first step in this direction, and consider the aspect of error control as future work.

## 4 Exploration with 20 Questions

To create a simple synthetic setting for validating our framework, we explore the properties of IP and C-IP by playing 20Q with two LLMs, one *Querier* LLM and one *Answerer* LLM. The task is simple: The Answerer LLM thinks of an animal and the Querier LLM is tasked to guess it within 20 queries.

### 4.1 Experiment Setup

**Dataset.** We obtain 20 common animal names from the Animals with Attributes 2 (AwA2) [136] dataset. On the one hand, this ensures that the LLMs we consider have sufficient knowledge about each class, allowing us to accurately evaluate our algorithm in a setting where the LLM performance is high. On the other hand, the AwA2 dataset provides an annotated set of binary attributes about each animal. This allows us to obtain a binary query set with expert-labeled answers. We consider a closed query set $\mathcal{Q}_{\text{closed}}$, in which 85 annotated attributes from AwA2 are converted into textual questions, and an open query set $\mathcal{Q}_{\text{open}}$, where queries are free-form questions obtained by prompting LLMs. We consider binary query-answers with $\mathcal{A}_{\text{binary}} = \{\text{"Yes", "No"}\}$ based on the AwA2 expert labels, and open query-answers with $\mathcal{A}_{\text{free-text}}$ being the set of English sentences (Examples in Appendix §J.1). In the 20Q setting, the data space is equal to the label space ($\mathcal{X} = \mathcal{Y}$), thus reducing the problem's complexity to help establish the effectiveness of using info-theoretic quantities in guiding LLMs.

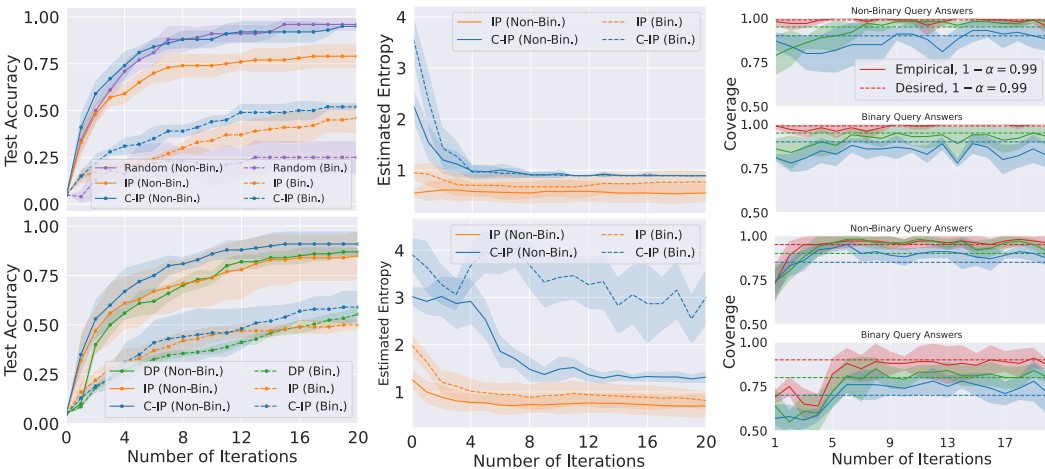

Figure 3: Evaluations of C-IP with a closed query set $\mathcal{Q}_{\text{closed}}$ *(top row)* and open query set $\mathcal{Q}_{\text{open}}$ *(bottom row)*. Each curve shows the average performance with shaded area as std.. *Left column*: Performance on the 20 Questions task of C-IP and baselines with binary (dashed) and non-binary (solid) query answers using Llama-3.1-8b. *Middle column*: Uncertainty estimated by IP and C-IP, with each curve evaluating the uncertainty of the selected query at each iteration. *Right column*: Desired (dashed) and empirical (solid) coverage of C-IP. Hyperparameter choice is marked with different colors, with desired and empirical coverage as dashed and solid.

**Models and Implementation.** We consider three open-source instruction-tuned models: Llama-3.1-8B [42], Qwen2.5-7B [7] and Phi-3-small [2]. We present results for Llama-3.1-8B in the main text, and other results in Appendix §E.1. To achieve the target coverage for both closed and open query sets, we calibrate using 100 random histories and labels. Every entropy term is estimated with 4 randomly drawn samples. We obtain our prediction sets $\mathcal{C}$'s by sampling via (13) for the setting with closed query set $\mathcal{C}_{\text{closed}}$ and via (13) for the setting with open query set $\mathcal{C}_{\text{open}}$. For LLM hyperparameters during generation, prompts, and further implementation details, refer to Appendix §G.

**Baselines.** To compare the performance of C-IP in the closed query set setting, we consider two baselines: *Random*, which uniformly selects a query from $\mathcal{Q}_{\text{closed}}$ at each iteration, and *IP*, which evaluates (2) at each iteration. For the open query set setting, the baselines are *Direct Prompting (DP)* and *Uncertainty of Thought (UoT)* [49]. Direct Prompting, as described in §3.2 and Appendix §I, directly prompts the LLM for a single query at each iteration. Furthermore, we compare C-IP against IP with calibrated probabilities obtained via Platt Scaling [92] and Temperature Scaling [46].

## 4.2 Results

**Predictive Performance.** We evaluate C-IP against baselines on predictive performance in Figure 3 (top left). For each iteration up to a maximum of 20, we evaluate the test accuracy given $q_{1:k}(x^{\text{obs}})$. We use a target coverage of $1 - \alpha = 0.9$. For the results with binary answers, we observe a sharp difference between random selection and IP-based methods, with C-IP performing better at each iteration. For non-binary answers, C-IP outperforms IP at every iteration, showing that early query selection weighs greatly for performance. While outperforming random selection in iterations 1-6, it performs comparably in iterations 7-20. We argue that this occurs due to the simplicity of the task, where the combination of open query-answers and random selection on task-relevant queries pre-determined by AwA2 provides sufficient information to achieve high accuracy. Overall, this result shows that C-IP is a better method for selecting queries from a pre-defined, closed query set.

We perform a similar performance evaluation for C-IP with $\mathcal{Q}_{\text{open}}$ in Figure 3 (bottom left). Here, we observe that C-IP with $\alpha = 0.15$ outperforms both baselines IP and DP. This shows that direct prompting may not always suggest the most informative query, and C-IP as an uncertainty estimate can potentially generalize to previously unseen queries.

**Uncertainty Estimation.** To ensure that C-IP selects informative queries, we evaluate the estimation of the entropy $\hat{H}(Y \mid q_{k+1}(X), q_{1:k}(x^{\text{obs}}))$ at each iteration. C-IP bounds this via the size of prediction sets in (11) and IP directly estimates it as in (2). In Figure 3 (top middle), we observe that, for both binary $\mathcal{A}_{\text{binary}}$ and free-text query answers $\mathcal{A}_{\text{free-text}}$, the C-IP bounds (blue curves) decrease as the algorithm progresses, whereas entropy estimates (orange curves) revolve around the same

level. This shows that C-IP uncertainty measures are more informative than entropy estimates. The results for C-IP with $\mathcal{Q}_{\text{open}}$ (Figure 3, bottom middle) also show a decrease in the estimated entropy as the number of iterations increases. In the case of binary query answers $\mathcal{A}_{\text{binary}}$, while we observe a less stable decay, C-IP still performs comparably with our baselines, showing that the estimation of uncertainty is robust despite noisy probabilities.

**Empirical Coverage.** To evaluate the coverage of our prediction sets, we choose three target values of $\alpha$ and evaluate the empirical probability that the true label is contained in the prediction set at each iteration over our test set (See Appendix §G.11 for empirical formula). Our results are shown in Figure 3 (right column). For $\mathcal{C}_{\text{closed}}$ with both binary $\mathcal{A}_{\text{binary}}$ and free-text query answers $\mathcal{A}_{\text{free-text}}$, the realized coverage is close to the target for $1 - \alpha = 0.99$, whereas our method undercovers for $1 - \alpha = 0.95$ and $0.99$. Overall, the choice of $\alpha$ largely influences the predictive performance of C-IP, which explains the superior performance for C-IP over IP and random query selection when $1 - \alpha = 0.99$. On the other hand, for $\mathcal{C}_{\text{open}}$ with both binary $\mathcal{A}_{\text{binary}}$ and free-text query answers $\mathcal{A}_{\text{free-text}}$, we observe a slight over-coverage for smaller choices of $\alpha$ and under-coverage for early iterations. This is likely due to the wide range of possible queries in few initial steps, leading to under-coverage and misspecification. See Appendix §F.1 for ablation studies regarding choices of $\alpha$.

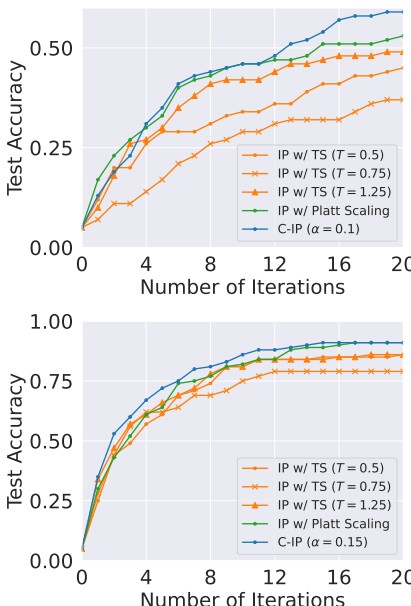

Figure 4: Comparison with Probability Calibration Baselines under open query set setting $\mathcal{Q}_{\text{open}}$ with binary *(top)* and free-text *(bottom)* query answers.

**Comparison with Probability Calibration Methods.** C-IP can be considered as a version of IP that can be used with miscalibrated measures of uncertainty. Here, we explore the option of first calibrating the predictive probabilities with popular calibration methods such as Platt Scaling and Temperature Scaling (TS), then applying IP with the updated probabilities. Briefly, Platt Scaling is a parametric method that learns a linear regression model using the calibration data to re-calibrate probabilities, and TS is a non-parametric method that divides the predicted probabilities by some temperature scalar $T$. We evaluate three choices of temperature $T \in \{0.5, 0.75, 1.25\}$. See more implementation details in §G.13.

The results of the comparison is shown in Figure 4. For an open query set $\mathcal{Q}_{\text{open}}$, C-IP outperforms IP with calibrated probabilities in the case of binary queries, and performs comparably with non-binary query answers. We hypothesize this occurs due to the complexity of the distribution of histories, where distribution-free uncertainty quantification methods such as SCP shine.

**Comparison with UoT.** We compare our method with a state-of-the-art LLM reasoning method, Uncertainty-of-Thought (UoT) [49]. UoT handles multi-turn conversations by propagating rewards based on information gain from future steps. At each iteration, it builds a shallow tree computing the information gain of a few potential trajectories, akin to seeing a few steps ahead. Importantly, UoT applies only to the setting with open query set and binary query answers. We will compare UoT with IP and C-IP, as both are entropy-based uncertainty quantification methods for interactive question answering. Further details are provided in Appendix G.

Our results are shown in Table 1. For binary query answers, UoT achieves higher success rate than IP or C-IP. This is reasonable as accumulating information gain a few iterations ahead is likely more advantageous than using immediate rewards. However, for free-text query answers, C-IP achieves a near 45% accuracy improvement, with a smaller average number of queries. Thus, using an immediate reward with free-text query answers works better than accumulated entropy-based rewards with binary answers.

Table 1: Comparison of C-IP with UoT reasoning, averaged across 5 runs with std..

| Method | Query Answers $\mathcal{A}$ | Avg. Len | Accuracy |
|---|---|---|---|
| UoT | binary | $13.57 \pm 1.69$ | $0.57 \pm 0.13$ |
| IP | binary | $11.40 \pm 0.78$ | $0.31 \pm 0.04$ |
| IP | free-text | $10.04 \pm 0.45$ | $0.65 \pm 0.18$ |
| C-IP ($\alpha = 0.1$) | binary | $11.87 \pm 0.83$ | $0.37 \pm 0.11$ |
| C-IP ($\alpha = 0.1$) | free-text | $10.58 \pm 0.58$ | $0.83 \pm 0.05$ |

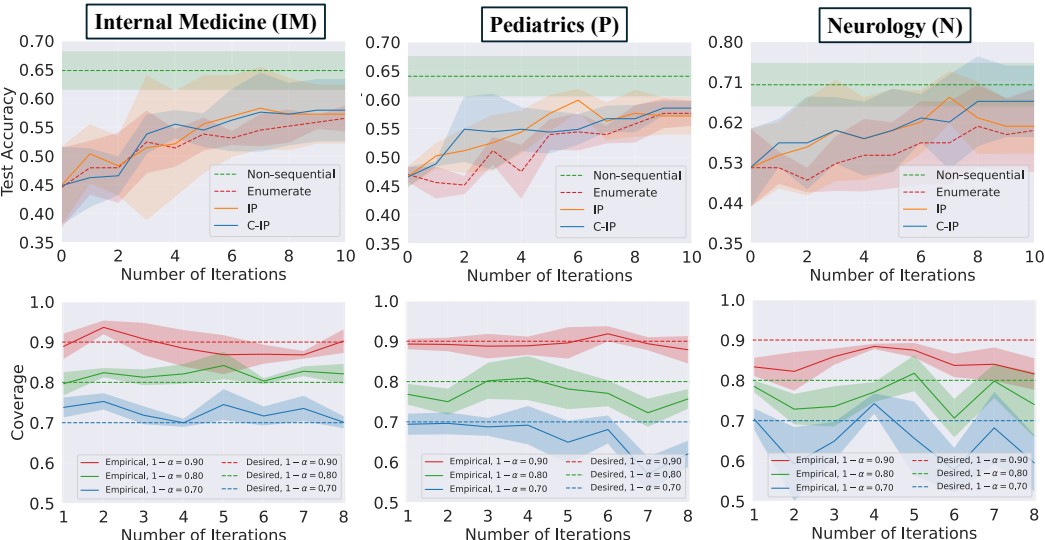

Figure 5: *Top:* Predictive Performance in Medical Interactive Question Answering on the MediQ dataset. The tasks are divided based on specialty. Desired coverage is $1 - \alpha = 0.8$ for IM and P and $1 - \alpha = 0.7$ for N. *Bottom:* Comparison of empirical and desired coverage for C-IP. Desired coverage (dashed curve) is shown for $1 - \alpha \in \{0.7, 0.8, 0.9\}$. Since the number of queries may depend on the datapoints, Empirical Coverage (solid curve) at iteration $k$ is evaluated for all test datapoints that have stopped at or before iteration $k$. Each curve is averaged over three splits, with the shaded area denoting their standard deviation.

## 5 Application to Interactive Medical Question Answering

In this section, we apply C-IP to an interactive medical question-answering task on the MediQ dataset. We provide the basic setup here and other details (such as prompts) can be found in Appendix §G.

### 5.1 Setup

**Dataset.** MediQ [70] is an interactive medical question-answering dataset extended from MedQA [53], a large-scale medical question-answering dataset designed to evaluate the professional knowledge and clinical decision-making abilities of LLMs. All data is drawn from medical licensing examinations in the United States. Each datapoint in MediQ contains a medical question, four multiple-choice answers, a label for the specialty/area, and a list of facts from a context paragraph about the patient[7]. For IP in this context, $\mathcal{X}$ is the set of patients' information, $\mathcal{Q}$ is the set of (closed) queries one can ask about the patient, and $\mathcal{Y}$ is the set of multiple-choice answers.

The task simulates an interactive setting between an Expert LLM (the predictor and the querier in our nomenclature) and a Patient LLM (the query-answerer), and the goal is that the Expert LLM diagnoses the patient by answering the provided medical questions. For our application, we consider medical questions from the following three categories: "Internal Medicine (IM)" (290 samples), "Pediatrics (P)" (217 samples) and "Neurology (N)" (108 samples). We treat each category as a single task, and the coverage is calculated using calibration data obtained from the same category.

**Models and Evaluation.** To ensure a fair evaluation, we divide each category into three equal sets: $\mathcal{D}_{\text{est}}$ for estimating entropy, $\mathcal{D}_{\text{cal}}$ for calibration, and $\mathcal{D}_{\text{test}}$ for test-set evaluation. We perform three-fold cross validation and evaluate the average performance and its standard deviation. In our main text, we evaluate Llama-3.1-8B as both the Expert LLM and the Patient LLM. Similar to 20Q, we first obtain the output token's logit score based on the four options (A, B, C, D), then apply softmax to obtain an estimate of the posterior distribution $\mathbb{P}(Y \mid q_{1:k}(x^{\text{obs}}))$.

**Baselines.** We compare with two baselines. In *Non-sequential* prediction, the Expert LLM makes a single prediction based on all necessary information. This serves as an upper bound on the predictive performance, and has been shown to outperform interactive methods [70]. In *Enumerate*, we provide the facts in the order of decomposition from the original context paragraph. Since our LLMs are inherently sequential, this acts as if the LLM is reading the context paragraph in order.

---

[7]The number of facts may vary depending on the number of sentences in each sample.

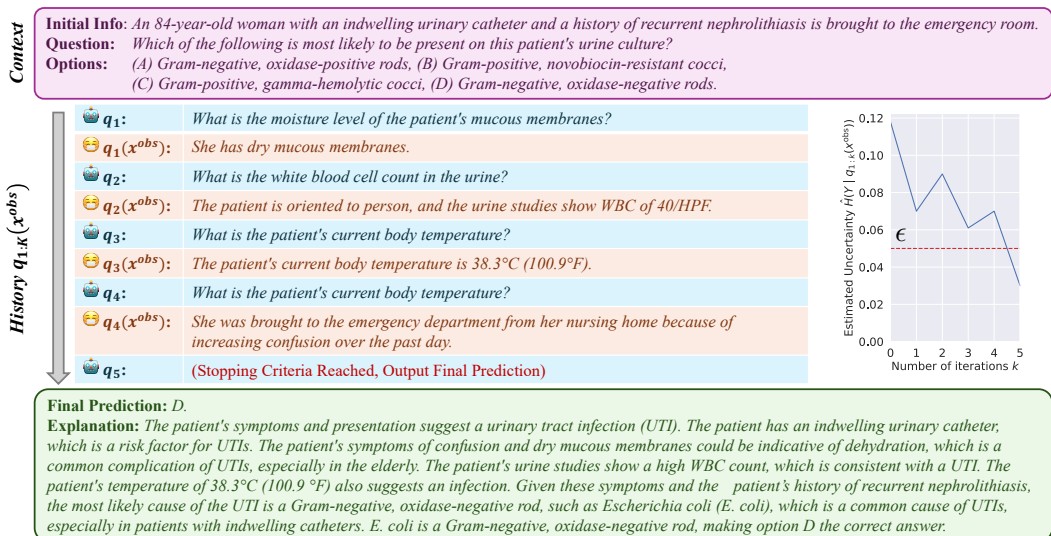

Figure 6: Example query-answer chain produced by C-IP for a patient in Internal Medicine. C-IP sequentially selects queries that approximately reduce uncertainty (i.e. average prediction set sizes from (11)) at each iteration. The posterior $\hat{P}(Y \mid q_{1:k}(x^{\text{obs}}))$ is also estimated and a prediction is made. Once uncertainty drops below a chosen threshold $\varepsilon$, the C-IP algorithm stops and makes a final prediction. Based on the obtained query-answer chain and the prediction, an explanation is also generated.

**Query Set Construction.** We find that constructing a single query set for the entire test set often leads to queries with unknown query answers. Hence, we construct a query set for each test datapoint based on the existing atomic facts in the input, by converting each fact into a question. Therefore, the goal of the IP algorithm is to select the queries with answers obtainable from the context. We find this setting more constructive compared to the setting with a single query set for all evaluations.

## 5.2 Results

**Predictive Performance and Coverage.** The performance of C-IP and the baselines for the three specialties is shown in Figure 5. C-IP and IP achieves comparable performance in all cases. This suggests that C-IP selects more informative queries during the interactive process. To explain the effectiveness of C-IP and why it did not strongly outperform baselines for N, we analyze their empirical coverage at each iteration. We observe that C-IP has valid coverage IM and P but undercovers for N. This is likely due to the small calibration set size for N (36 datapoints).

**Explanations.** We show a run-down of the C-IP algorithm in Figure 6. Before any query selection, the Expert LLM is provided with initial information, the medical question, and possible options for the answer. Then, C-IP selects the first query $q_1 = $ "What is the moisture level of the patient's mucous membranes?", to which the Patient LLM responds $q_1(x^{\text{obs}}) = $ "She has dry mucous membranes". This process continues until the uncertainty measured based on the posterior of the answer options from the Expert LLM drops below 0.01. Once C-IP stops, it makes the final prediction "D", and summarizes the query-answer chain into a paragraph explaining the decision. More examples (including other specialties) are shown in Appendix §J.2.

## 6 Conclusion and Future Work

In this work, we proposed C-IP, a novel sequential information pursuit algorithm that uses prediction set sizes as an alternative to conditional entropy to estimate uncertainty. We leveraged a mathematical connection between prediction sets and entropy, and proposed two sampling methods for obtaining histories. Through our empirical studies on 20Q and MediQ datasets, we demonstrated that our proposed approach performs well in practice, and that the size of the prediction sets is an effective measure of uncertainty. As we suggested in §3 Remark 2, an interesting avenue for future work is to provide guarantees for the correctness of the prediction given query-answer chains, $\mathbb{P}(Y \mid q_{1:k}(x^{\text{obs}}))$, obtained by C-IP and for the query-answer chains $q_{1:k}(x^{\text{obs}})$ themselves. Doing so would require a careful characterization of the distribution $Q_{1:k} \sim \text{IP}(\{q_i, q_i(X)\}_{1:k})$.

## Acknowledgements

K. H. R. C. would personally like to thank Tianjiao Ding, Uday Kiran Reddy Tadipatri, Liangzu Peng, Darshan Thaker, Kyle Poe, Buyun Liang, Ryan Pilgrim, Aditya Chattopadhyay, and Zitong Yang for fruitful discussions. K. H. R. C., Y. G., and R. V. acknowledge the support from NSF 2031985, Simons Foundation 814201 (THEORINET), University of Pennsylvania Startup Funds, NSF Graduate Research Fellowship Program, Penn Engineering Dean's Fellowship and Amazon Fellowship. E. D. was partially supported by the NSF, ARO, AFOSR, ONR, and the Sloan Foundation. H. H. was supported by the NSF CAREER award CIF-1943064.

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

# Appendix

## A  Outline

## B  Related Work

### B.1  Large Language Models and Interactive Settings

**Information-Seeking Environments of LLMs.** In information seeking environments, LLM agents seek information by constructing well-design prompts and designing proper rewards with reinforcement learning (RL) to encourage proper information-seeking behavior. Prompting-based methods [1, 87, 126, 126, 127, 134] typically involve LLM agents interacting and solving a task together. Advanced techniques include building graphs [52], using tools such as APIs and databases [96, 137], and leveraging Chain-of-Thought (CoT) methods [22, 49, 75, 127] to ensure interaction is valid and improve performance. However, they do not explicit attempt to minimize the average number of interactions needed. On the other hand, RL-based methods such as ReAct and DeLLMa [76, 139] usually involves designing a good reward or utility function (can be non-entropy-based) that balances between exploration and exploitation, which our method can be classified in the pure exploitation regime. By and large, prompting-based methods and RL-based methods consider a different strategy that is not as explicit as our work here, and are not always plug-and-play to the interactive question-answering setting.

**Interactive Medical Question Answering Setting.** As our motivation (see Fig. 1), interactive medical question answering is one potentially impactful application of our work. While there are many existing benchmarks of medical question answering on LLMs such as MedQA [53], MedQuAD [10], MedMCQA [85], PubMedQA [54], they mostly focus on single-turn one-input-one-response type interaction. Instead, we focus on a more recently trending setting of interactive medical question answering and diagnosis. Datasets in this setting are still rare, with the two being the WMeDiaQA [115], which is a chinese interactive dataset, and the MediQ [70]. Importantly furthermore, it has been shown that LLMs in interactive human-LLM medical/clinical settings are subpar: as shown in Bean et al. [9], they perform nearly 30% worse than LLMs acting alone. Similarly, Li et al. [70] has also shown that directly applying LLMs for interactive settings perform worse without additional guidance to the iterative process.

To summarize, the research question of why LLMs in a multi-turn, interactive environments perform worse than single-turn use-cases remains active and highly relevant in today's use-cases [63, 70]. Our work seeks to provide some answer via the lens of information gain.

### B.2  Uncertainty Quantification

**Predictive Inference and Conformal Prediction.** Our proposed formulation uses entropy and mutual information, which are fundamental measurements of uncertainty that dates back to Shannon's Information Theory [79, 107, 108] and predictive inference [103, 119, 120, 125, 133]. Amongst existing methods for predictive inference, conformal prediction has gained popularity due to its flexibility for constructing prediction intervals with a marginal coverage guarantee under the exchangeability

of data points. While distribution-free inference and the conformal prediction framework have been extensively studied in recent works, with many going beyond standard assumptions [see, e.g., 5, 8, 25, 33, 34, 43–45, 56, 65–69, 72, 73, 88–91, 93, 100, 102, 105, 110, 122–124], few has formalized the framework in the interactive and iterative setting. Arguably, the most similar setting is the online or adaptive setting [35, 36, 40, 41, 116, 130, 135], where distribution shift exists and the exchangeability assumption no longer holds. This line of research is different from our work in two major ways: 1) they focus on an online setting where data points are sequentially provided, such as time series, with assumptions such as affine transformations for the distribution shifts. In contrast, our work focuses on an interactive question answering setting that depends on multiple factors, such as the query set $\mathcal{Q}$ (open and closed setting), the query answers $q_{1:k}(x^{\mathrm{obs}})$, and the target distribution $Y$; 2) they focus on providing tighter risk control in an online setting, whereas our work focuses on how to use uncertainty measured from prediction sets to drive decision making. While we agree adaptive conformal prediction and sequential information gain via conformal prediction draw some resemblances, we reserve the research of their precise relationship in a future work.

**Uncertainty Quantification for LLMs.** Methods for quantifying uncertainty in LLMs largely revolve around controlling the quality and ensuring correctness of LLM generations, either via minimizing entropy with a constructed distribution [13, 20, 48, 74, 86, 112, 141] or leveraging conformal prediction [24, 50, 57, 62, 81, 94, 97, 104, 106, 114, 117, 129, 131, 141, 146]. Here we highlight the ones that are highly relevant to our method: Ling et al. [75] considers the problem of uncertainty quantification in the context of in-context learning. Similar to our IP baseline, it also measures uncertainty by computing the entropy of distributions estimated from the token distribution. On the other hand, Kumar et al. [61] applies conformal prediction to question answering with multiple choices. Their method also extracts probability of each choice by obtaining the logits of each answer token, but their setting is single-turn rather than multi-turn. Last but not least, Chang [15] considers multi-turn LLM collaboration debates while using mutual information as a measurement of dialogue and diversity. As aforementioned, there exist multiple methods to measure uncertainty in LLMs, and we reserve how different choices might affect the interactive question-answering process in this paper's setting for future work.

**Uncertainty Quantification in Interactive Environments.** The two closely related works are Uncertainty of Thought (UoT) [49] and EQA work by Ren et al. [98]. UoT is in the setting of efficient reasoning for LLMs, in which the method formulates its exploration as a tree. At each iteration, UoT explores multiple branches of solutions with up to a depth at three, then evaluates the accumulated information gain at the leaf node, ultimately choosing the leaf with the highest information gain. Compared to C-IP, UoT considers only binary query-answers, considers accumulated rewards, and computes entropy directly, whereas C-IP considers free-text query answers, immediate rewards at each iteration, and utilizes calibrated information. On the other hand, Ren et al. [98] is positioned the setting of robotics, where the Vision-Language Model (VLM) agent has to answer a question by exploring different parts of the room. While this work also uses conformal inference and prediction set size to estimate confidence, the connection to entropy was not presented. It also differs from our language-domain setting by using a custom, domain-specific score function called "relevance" rather than the probability score as in the case of C-IP.

### B.3 Sequential Information Gain via Information Pursuit

**Previous Iterations of IP.** Information Pursuit framework was first proposed by Jahangiri et al. [51] as an active-testing algorithm for scene parsing. Later learning-based approaches, such as generative approach [16] and variational approaches [17, 19, 26], require having access to the data distribution and learning a model to estimate the posterior distribution. Our work differs from all of the above in multiple aspects: 1) learning-based approaches of IP largely focus on computer vision tasks such as image classification and image generation, whereas our work focuses on the language domain; 2) their setting often provides sufficient samples that allows for a good estimation of the posterior distribution $\mathbb{P}(Y \mid q_{1:k}(x^{\mathrm{obs}}))$, which enables efficient implementation with faster and scalable inference and make it applicable to large-scale tasks. In contrast, LLMs may not provide good estimates of the posterior distribution because they are pretrained models; 3) IP has mostly been written as an interpretability framework rather than a framework for interactive guidance. It's unclear yet how IP will perform in settings where multiple off-the-shelf models are involved; 4) query set in previous iterations of IP is always a pre-determined fixed set, whereas our work explores explores both the closed setting and an open generation setting where queries are obtained from

LLMs iteratively; and 5) query answers in previous iterations of IP are either concept-based or raw features such as pixel-values. All the reasons above provide strong motivation to our current work.

**Efficient Sampling.** The challenge of efficiently obtaining samples for estimating mutual information often arises in IP. The generative approach [16] learns a joint distribution of $\mathbb{P}(\mathcal{Q}(X), Y)$, then uses Langevin Dynamics to compute the conditional distribution. Alternative, V-IP [17], the current state-of-the-art implementation of IP, learns a query selection function $g$ parameterized by a neural network $g_\theta$, which also requires efficient sampling of random histories. In this work, our uniform parameterization is inspired from V-IP, whereas our sampling via LLM prompting provides a new domain-specific way to sample query-answer chains. This technique is also applied to extensions of IP for other applications [19, 37, 60].

## C Limitations

Our work explores the areas of using LLMs interactively and using information gain to obtain further information. While efficient, it still requires sampling in order to estimate quantities such as conditional entropy and prediction sets. One potential exploration is to find ways to estimate posteriors via a single pass, for example, via tractable approximations. Moreover, the success of our method also relies on LLMs that can follow instructions well, which today's LLMs are still not fully reliable yet and instruction-following in general is still an active area of research. Furthermore, as stated in our Section 6, we are not fully leveraging the potential of conformal prediction as we are only using it as a measure of uncertainty but not using it as means to control risk and evaluate the correctness of the overall query-answer chains. We reserve these interesting directions for future work.

## D Broader Impacts

Our work studies the setting of using LLMs interactively, where information is not provided all at once but sequentially depending on the previous responses of LLMs. While currently understudied, this research direction aligns closely with how human-human or human-LLM behaves in the real world, and potentially describes the predominant way of how human would use LLMs as the capabilities of LLMs grows. Hence, it is crucial to understand the behavior through careful mathematical formulation and experimental designs. We argue that the formulation in this work can serve as the foundational ground work for further analysis when interactively using LLMs. Our empirical study with interactive medical question answering also demonstrates potential applications of our method in real-world medical settings.

# E Additional Results

## E.1 20Q

The predictive performance of 20Q on two additional models (Qwen2.5-7B [7] and Phi-3-small [2]) are shown in Figure 7 and Figure 8, respectively. Moreover, we visualize the different thresholds obtained from split conformal prediction in Figure 9, with thresholds for each model in each row. We will discuss the performance of C-IP with respect to the thresholds obtained together, as they demonstrate a picture of when C-IP would be successful and unsuccessful in guiding LLMs.

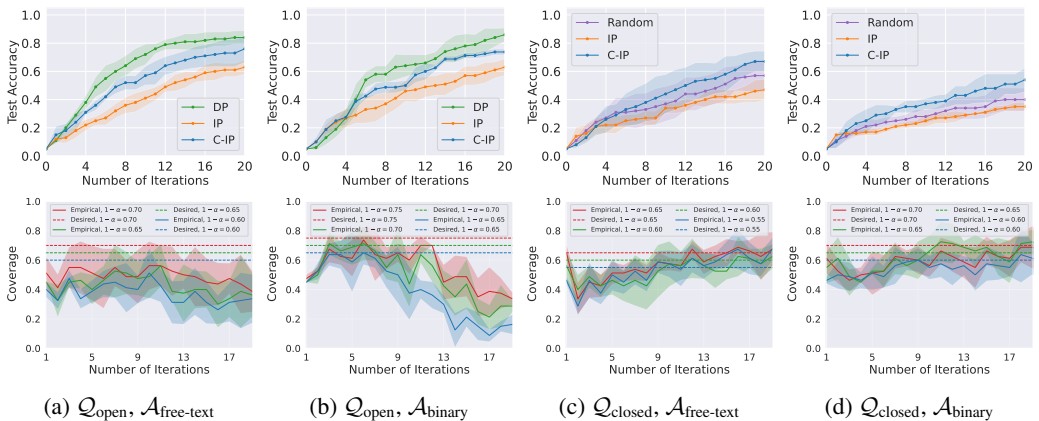

(a) $\mathcal{Q}_{\text{open}}, \mathcal{A}_{\text{free-text}}$     (b) $\mathcal{Q}_{\text{open}}, \mathcal{A}_{\text{binary}}$     (c) $\mathcal{Q}_{\text{closed}}, \mathcal{A}_{\text{free-text}}$     (d) $\mathcal{Q}_{\text{closed}}, \mathcal{A}_{\text{binary}}$

Figure 7: Predictive Performance of 20Q with Qwen-2.5-7B model.

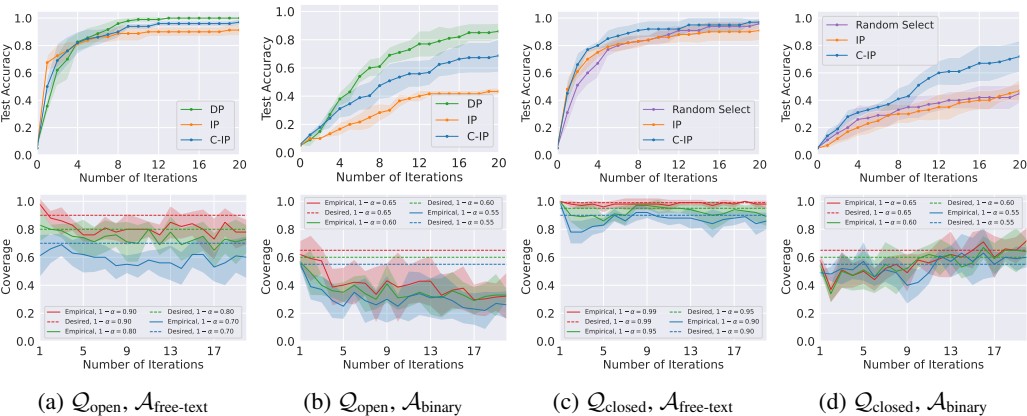

(a) $\mathcal{Q}_{\text{open}}, \mathcal{A}_{\text{free-text}}$     (b) $\mathcal{Q}_{\text{open}}, \mathcal{A}_{\text{binary}}$     (c) $\mathcal{Q}_{\text{closed}}, \mathcal{A}_{\text{free-text}}$     (d) $\mathcal{Q}_{\text{closed}}, \mathcal{A}_{\text{binary}}$

Figure 8: Predictive Performance of 20Q with Phi-3-small model.

**Successful Cases of C-IP.** We first discuss the setting of 20Q with closed query set $\mathcal{Q}_{\text{closed}}$. It is clear that for all three models (including Llama-3.1-8B from Figure 3), C-IP outperforms IP and random query selection, where it is able to achieve a higher accuracy at every iteration. One sign of this success can be attributed to the fact that our empirical marginal coverage guarantee follows closely to the desired guarantee at each iteration.

Alternatively, one can attribute this success by looking at the thresholds obtained through split conformal prediction from Figure 9. In the closed query set setting (right two columns), we observe a progressive increase in the thresholds also as the number of iterations increase. This matches with our intuition: in the first few iterations, the model is not confident about its prediction, hence the threshold $\tau$ remains small, and a large number of classes $y$ would be in the prediction set. As the number of iterations increase, the predictor (on average relative to the calibration set) becomes more confident, hence the threshold $\tau$ increases.

**Unsuccessful Cases of C-IP.** We now turn to the unsuccessful cases of C-IP, which largely pertains the open query set setting for Qwen2.5-7B and Phi-3-small. We observe that C-IP in fact does not

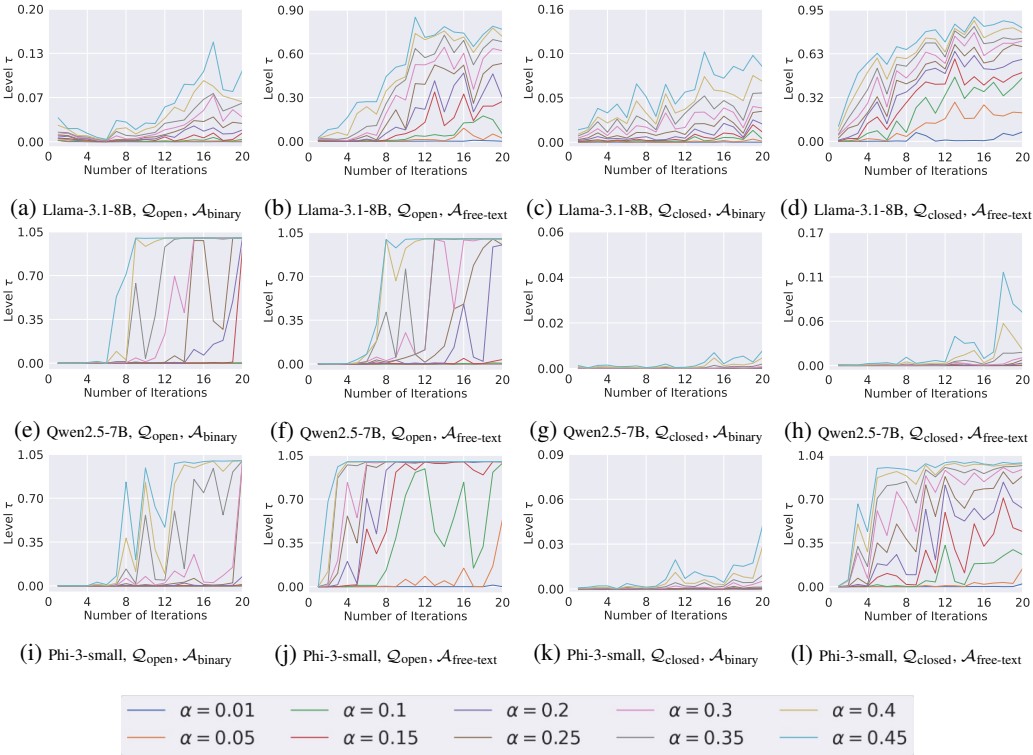

Figure 9: Thresholds obtained through split conformal prediction for 20Q. The legend below shows different colors Each color corresponds to different choices of target coverage $\alpha$.

outperform the DP method. To understand why, we argue it from the perspective of *meaningful calibration*.

Recall that in the open query set setting, C-IP is calibrated on query-answer chains obtained from DP. Turning to Figure 7 (a, b) and Figure 8 (a, b), we observe that the empirical coverage is below the desired coverage a majority of the iterations, a sharp contrast to successful cases of C-IP where empirical and desired coverage is mostly coincides. To explain the misalignment between the empirical coverage and desired coverage, we can also infer from Figure 9, where we observe sudden jumsp in the thresholds in the open query set setting for Qwen2.5-7B and Phi-3-small. From our results, we observe the following: Given Qwen2.5-7B and Phi-3-small models are relatively strong models and our task is relatively simple, it is often the case where the label is guessed correctly within a few iteration. Once the correct prediction is made, it becomes part of the history $q_{1:k}(x^{\mathrm{obs}})$ (e.g. "Is the animal a dolphin? Yes.") and any new query no longer provides any meaningful information gain. This ultimately leads to an over-confident prediction for every data point, yielding an uninformative threshold. As a result, every prediction now requires a very high probability (near 1) to be an element in the prediction set. Consequently, the prediction sets cannot quantify uncertainty in a meaningful way, leading to suboptimal results.

While there exist cases where C-IP does not outperform other methods, we argue it is more interpretable than IP in that one can observe its failure modes from the thresholds obtained and empirical coverage. As discussed in our Future Work (Section §6) and Limitations (Appendix §C), this direction of obtaining the *proper* conformal coverage guarantee over distributions of natural language deserves further study, and we reserve this for future work.

**Additional Comparisons with UoT.** We provide the full comparison with UoT for Llama-3.1-8B in Table 2. Focusing on IP and C-IP, we can see that C-IP is able to outperform IP in all four settings, which aligns with our previous results regarding the predictive performance. Comparing C-IP with UoT, C-IP is able to outperform UoT in free-text query answers, but not in closed binary query answer setting. This can be explained by the design of the respective methods: C-IP relies on immediate reward (information gain) whereas UoT looks ahead, accumulates and propagates rewards up to the certain depth before selecting the next query.

As discussed previously in Section §E.1, while having binary query answers allow UoT to look forward and accumulate rewards (akin to a beam search), free-text query answers provide a much larger marginal gain over binary query answers and is a more realistic setting to consider.

Table 2: Full Comparison of C-IP with UoT reasoning, averaged across 5 runs with shaded area as standard deviation.

| Method | Query Set $\mathcal{Q}$ | Query Answers $\mathcal{A}$ | Avg. Len. | Avg. Success Len. | Success Rate |
|---|---|---|---|---|---|
| UoT | open | binary | $13.57 \pm 1.69$ | $8.75 \pm 0.82$ | $0.57 \pm 0.13$ |
| IP | open | binary | $11.40 \pm 0.78$ | $10.48 \pm 0.98$ | $0.31 \pm 0.04$ |
| IP | open | free-text | $10.04 \pm 0.45$ | $9.66 \pm 0.68$ | $0.65 \pm 0.18$ |
| IP | closed | binary | $10.74 \pm 0.59$ | $10.98 \pm 0.86$ | $0.29 \pm 0.11$ |
| IP | closed | free-text | $13.81 \pm 0.16$ | $13.73 \pm 0.24$ | $0.71 \pm 0.08$ |
| C-IP ($\alpha = 0.1$) | open | binary | $11.87 \pm 0.83$ | $12.12 \pm 0.26$ | $0.37 \pm 0.11$ |
| C-IP ($\alpha = 0.1$) | open | free-text | $10.58 \pm 0.58$ | $10.45 \pm 0.63$ | $0.83 \pm 0.05$ |
| C-IP ($\alpha = 0.1$) | closed | binary | $11.26 \pm 0.84$ | $10.58 \pm 0.61$ | $0.29 \pm 0.04$ |
| C-IP ($\alpha = 0.1$) | closed | free-text | $8.87 \pm 0.31$ | $8.24 \pm 0.51$ | $0.83 \pm 0.07$ |

## E.2 MediQ

The thresholds $\tau$ obtained through split conformal prediction is shown in Figure 10. Similar to our observations in the 20Q case, we observe that a useful and successful case of C-IP can be attributed to having good calibration and finding meaningful thresholds.

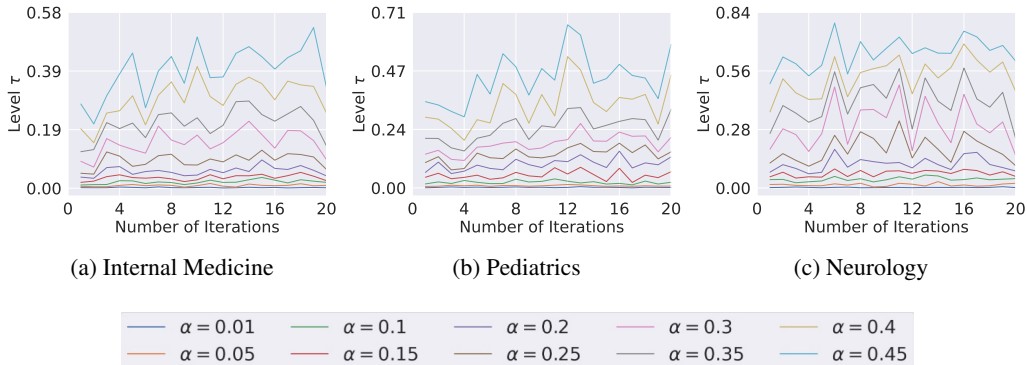

(a) Internal Medicine      (b) Pediatrics      (c) Neurology

| | $\alpha = 0.01$ | | $\alpha = 0.1$ | | $\alpha = 0.2$ | | $\alpha = 0.3$ | | $\alpha = 0.4$ |
| | $\alpha = 0.05$ | | $\alpha = 0.15$ | | $\alpha = 0.25$ | | $\alpha = 0.35$ | | $\alpha = 0.45$ |

Figure 10: Thresholds obtained through split conformal prediction for MediQ. The legend below shows different colors Each color corresponds to different choices of target coverage $\alpha$.

# F  Ablation Studies

## F.1  Varying Desired Coverage

We compare predictive performance of 20Q (Figure 11) and MediQ (Figure 12). In nearly all cases, we observe only minor differences between different levels of $\alpha$, which indicates C-IP is fairly robust with the appropriate choices of $\alpha$.

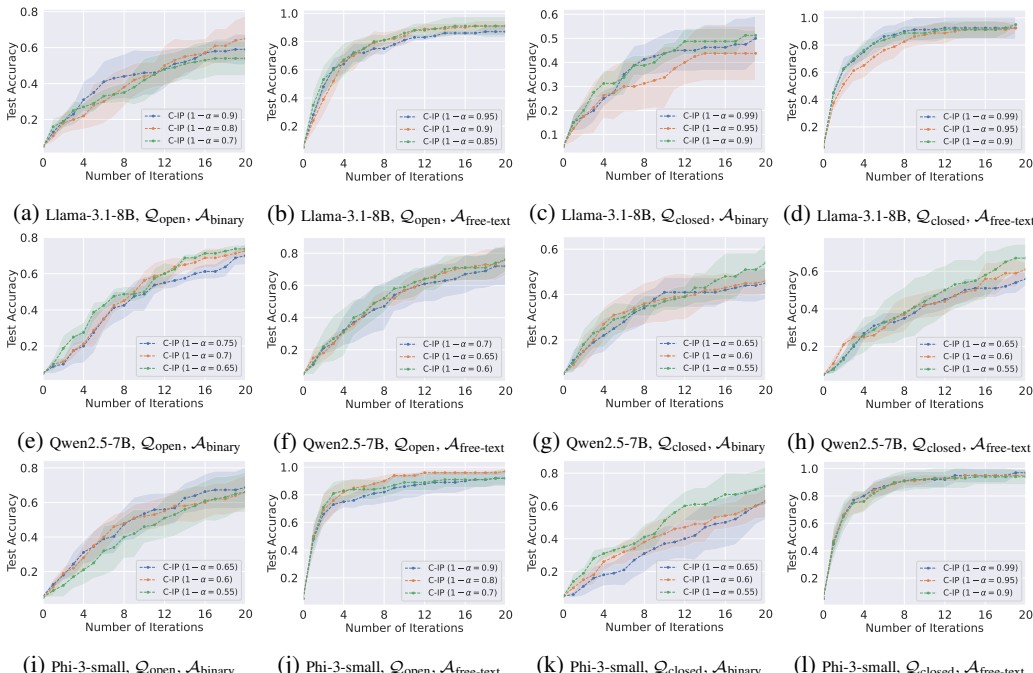

Figure 11: Predictive Performance of 20Q with different choices of $\alpha$.

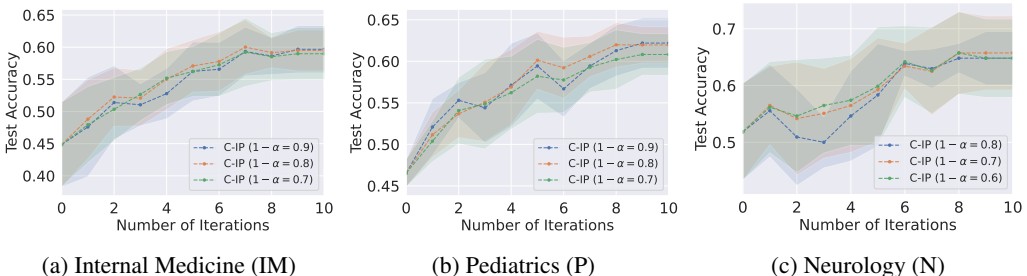

Figure 12: Predictive Performance of MediQ with different choices of $\alpha$.

## F.2  Varying Number of Estimation Samples

We evaluate whether C-IP is sensitive to the number of samples $n_{est}$ used during estimation for the task of 20Q on Llama-3.1-8B (Figure 13). Recall that by the number of estimation samples $n_{est}$, we refer to the number of samples used to estimate the upper bound (C-IP). In our ablation study for 20Q on Llama-3.1-8b with closed query set $\mathcal{Q}_{closed}$, binary query answers $\mathcal{A}_{binary}$ and open query set $\mathcal{Q}_{open}$, binary query answers $\mathcal{A}_{binary}$, we find that the number of samples do have impact on the performance. While one would expect the more is better, but we find that using $n_{est} \in \{1, 2, 4, 8\}$ performs somewhat similarly. One possible explanation that aligns with empirical observations of LLMs is that stochasticity plays a role in obtaining better predictive performance in LLMs, akin to how LLMs decoding via sampling based on output token distribution often outperforms greedy decoding where tokens are selected deterministically by choosing the most likely token [47].

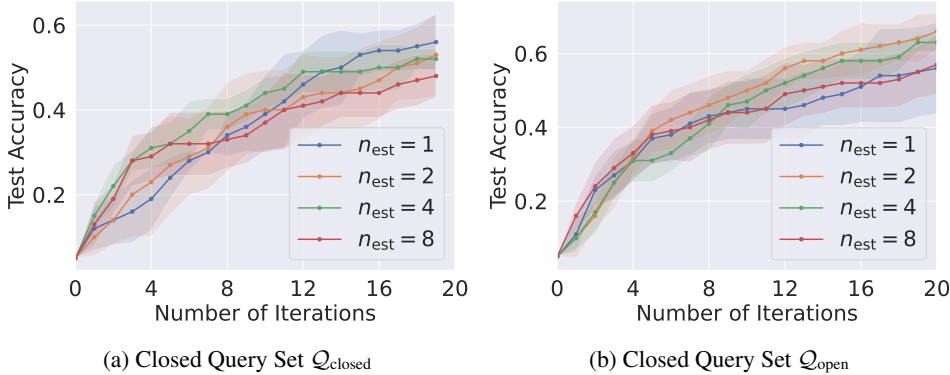

(a) Closed Query Set $\mathcal{Q}_{\text{closed}}$          (b) Closed Query Set $\mathcal{Q}_{\text{open}}$

Figure 13: Performance of 20Q with different number of estimation samples. Each curve corresponds to the averaged accuracy over 5 runs, with shaded areas representing their standard deviation.

### F.3 Varying Number of Queries Sampled for Open Query Set.

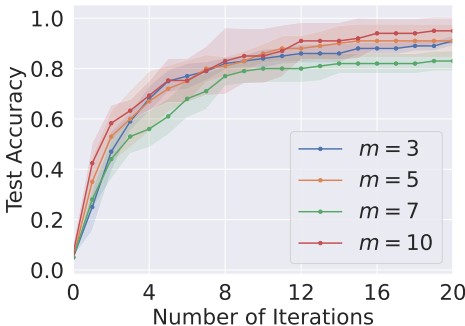

Figure 14: Performance of 20Q in the open query setting with different number of sampled queries $m$. Here we focus on Free-text Query Answer $\mathcal{A}_{\text{free-text}}$. Each curve corresponds to the averaged accuracy over 5 runs, with shaded areas representing their standard deviation.

We evaluate whether C-IP on Llama-3.1-8B for 20Q with the open query set $\mathcal{Q}_{\text{open}}$ is sensitive to the number of queries $m$ asked at each iteration (Figure 14). We observe a comparable performance for all choices of $m$.

# G Implementation Details

## G.1 Reproducibility

Our code is available at

https://www.github.com/ryanchankh/ConformalInformationPursuit/.

## G.2 Computational Resources

The experiments are conducted on a workstation of 8 NVIDIA A5000 GPUs. Each LLM used in this work is able to load and run on a single A5000 GPU.

## G.3 Code

All experiments are implemented in Python 3.12. The main packages used are hugging-face, PyTorch, Numpy, and TogetherAI API (for UoT baseline). The three models used are `meta-llama/Meta-Llama-3.1-8B-Instruct`, `microsoft/Phi-3-small-128k-instruct`, and `Qwen/Qwen2.5-7B-Instruct`. Unless stated otherwise, we use the default hyperparameters from huggingface. We use the following LLM hyperparameters every time we inference: `do_sample=True`, `temperature=0.7`, and `max_new_tokens=1024`.

## G.4 Classes for 20Q and MediQ

For 20Q, we select 20 classes from all 50 classes that are available in the Animals with Attributes 2 [136] dataset. The 20 classes are: giraffe, zebra, elephant, killer whale, dalmatian, polar bear, giant panda, hippopotamus, rhinoceros, lion, tiger, blue whale, walrus, grizzly bear, siamese cat, cow, german shepherd, gorilla, dolphin, and moose.

For MediQ, each medical question is a multiple-choice question with four options: A, B, C, and D. Note that the possible choices and order of the options are different for each problem.

## G.5 Obtaining Probabilities from LLM

In this work, we focus on LLMs where logit scores of output tokens are accessible. While there are many methods for obtaining a posterior distribution $\mathbb{P}(Y \mid q_{1:k}(x^{\text{obs}}))$ for a given $x$, we estimate the posterior by first obtaining the LLMs' output token logits based on the class labels' tokens, then applying softmax function. When certain class names consist of multiple tokens, we select the first token's probability to represent the class's probability. For all of our models, none of the classes consist of the same *first* token. In the case that happens, one may consider adding a prefix such as enumerations ("1.", "2.", ...) or leveraging special symbols trained with large language models.

## G.6 Entropy Estimation

Estimation of terms such as entropy in (IP) or upper bounds in (C-IP) requires samples of $(X, Y)$ pairs. We use 4 randomly drawn samples for all 20Q experiments whenever a entropy term needs to be estimated and use 12 randomly drawn samples from the estimation set $\mathcal{D}_{\text{est}}$ for all MediQ experiments.

To be specific, the empirical estimation $\hat{H}(Y \mid q_{1:k}(x))$ with a probability estimator $\hat{P}(Y \mid q_{1:k}(x)) := f(q_{1:k}(x))$ is estimated by

$$\hat{H}(Y \mid q_{1:k}(x)) = \sum_{(x,y) \in \mathcal{D}_{\text{cal}}} f(q_{1:k}(x)) \log_2 f(q_{1:k}(x)) \tag{15}$$

## G.7 Calibration

Achieving marginal coverage in (12) requires calibration samples. For 20Q, we sample 100 randomly drawn labels. For MediQ, we sample 200 randomly drawn datapoints with replacement in the calibration set $\mathcal{D}_{\text{cal}}$.

### G.8 Preprocessing Query Answers for 20Q

To avoid generating query-answers from LLMs at each iteration for every query, in the case of closed query set $\mathcal{Q}_{\text{closed}}$ and free-text query-answers, we generate query-answers offline by sampling 10 possible query-answers for every class and query. Then, in our experiments, for any given query and class, we select a query answer by uniformly sampling one out of the ten generated answers. For binary answers, we only consider query-answers with "Yes" and "No", without any variability.

### G.9 Evaluation of Predictive Accuracy

We discuss the two evaluation methods for 20Q and MediQ separately as they are different. 20Q is often played in a manner that the Querier LLM keeps guessing until either it guesses correctly or until the maximum number of iterations $L$ is reached[8]. Here, we following the same protocol. If the Querier LLM guesses the correct prediction $\hat{y}_k^*$ at iteration $k$, then $\hat{y}_k = \hat{y}_i^*$ for $i = k + 1, \ldots, L$. For MediQ, since the neither the Expert LLM or Patient LLM knows what the correct prediction is, we require C-IP to reach the stopping criteria (See Appendix §G.10). For any given datapoint $x^{\text{obs}}$, suppose the stopping criterion is reached at iteration $\tilde{k} < L$ and $\hat{y}_{\tilde{k}}$ is the prediction. Then we assume the prediction does not change after iteration $\tilde{k}$ and set $\hat{y}_i = \hat{y}_{\tilde{k}}$ for $i = \tilde{k} + 1, \ldots, L$.

### G.10 Stopping Criteria

As mentioned in Section 3.2 Remark 1, arriving at $I(q(X); Y \mid q_{1:k}(x^{\text{obs}})) = 0$ in practice is highly unlikely. Here, we explain how one goes about arrive at an approximation $I(q(X); Y \mid q_{1:k}(x^{\text{obs}})) \approx 0$.

One approach is to utilize the model confidence $\mathbb{P}(Y \mid q_{1:k}(x^{\text{obs}}))$. In previous variations of IP, including the generative version [16] and the variational approaches [17, 29], the algorithm stops querying once the posterior $\mathbb{P}(y \mid q_{1:k}(x^{\text{obs}})) > \varepsilon$ for any $y \in \mathcal{Y}$. This is suited for previous cases because the predictor model $f$ is learned from training data, which provides better estimates for the posterior. Unfortunately, in the case where model confidence is miscalibrated, this process becomes unstable and unreliable.

Empirically, we observe that while estimated conditional entropy (either with direct computation or with approximation via prediction sets) for every $q$, i.e. $\hat{H}(Y \mid q(X), q_{1:k}(x^{\text{obs}}))$, does not converge to 0 as the sequence length increases. However, it tends to converge to some constant number when the algorithm is confident enough to some number. Hence we propose to calculate the standard deviation between obtained estimations. Precisely, given a query set $\mathcal{Q}$ (closed or open), the query-answer chain $q_{1:k}(x^{\text{obs}})$, and a stopping threshold $\varepsilon$, the stopping criterion can be described as:

- *Step 1.* Let
$$c_i = \hat{H}(Y \mid q(X), q_{1:k}(x^{\text{obs}})) \quad \text{for } q \in \mathcal{Q}. \tag{16}$$

- *Step 2.* Compute the estimated standard deviation
$$\bar{c} = \sum_{i=1}^{|\mathcal{Q}|} c_i, \quad \hat{\sigma} = \sqrt{\frac{1}{|\mathcal{Q}|} \sum_{i=1}^{|\mathcal{Q}|} (c_i - \bar{c})^2}. \tag{17}$$

- *Step 3.* Stop if $\hat{\sigma} < \varepsilon$, else continue the algorithm.

### G.11 Evaluation of Empirical Coverage

Suppose we have a test set $\mathcal{D}_{\text{test}} = \{(x_i, y_i)\}_{i=1}^{n_{\text{test}}}$ with size $n_{\text{test}}$ for evaluation. The empirical coverage at iteration $k$ is evaluated by

$$\mathbb{P}_{X,Y,Q_{1:k},\mathcal{D}_{\text{cal}}} (Y \in \mathcal{C}_{\hat{\tau}_k}(Q_{1:k}(X))) \approx \frac{1}{n_{\text{test}}} \sum_{i=1}^{n_{\text{test}}} \mathbb{1}\{y_i \in \mathcal{C}_{\hat{\tau}_k}(q_{1:k}(x_i))\}, \tag{18}$$

where $(x_i, y_i) \in \mathcal{D}_{\text{test}}$ is the $i$-th test sample, $\mathcal{C}_{\hat{\tau}_k}$ is the prediction set for query-answer chains of length $k$, $q_{1:k}(x_i)$ is the obtained query answer chain (e.g. from C-IP), and $\mathbb{1}(\cdot)$ is a binary indicator function, which equals to 1 when the condition in the parameter is true and 0 otherwise.

---

[8]This is also consistent with UoT [49].

### G.12 Hyperparameters for UoT Baseline

Unless specified below, we follow the default settings as suggested in UoT's main manuscript [49] and their Github repository https://github.com/zhiyuanhubj/UoT/tree/main/src/uot.

For results in Table 1, to ensure the fairness of the experiments, we set the number of branches in their method (`n_potential_actions` in their code) to 5. Their experimental results are produced with API from TogetherAI. Please see their website https://www.together.ai/ for more details. The total charges to produce our results for the three models cost no more than $50.

### G.13 Details on Probability Calibration Methods

For Platt Scaling, for each length $k \in \{1, \ldots, L\}$, we sample randomo histories of length $k$ from the calibration data $\mathcal{D}_{\text{cal}}$ and train a logistic regression for each class. The logistic classifier is implemented using Scikit-learn python package [14] using default hyparameters ($\ell - 2$-regularization and a inverse of regularization strength of 1).

For temperature scaling (TS), we did not do any hyperparameter search but reported the best three curves for $T \in \{0.5, 0.75, 1.25\}$.

# H Prompts

Here we provide the prompts used in our experiments.

## H.1 20Q

---

**System Instruction for Querier LLM**

```
You are an expert on animals. Your goal is to predict the animal given the information you have gathered. The
↪  animal must be one of the following: {class_names}. Be as precise and as direct as possible. You may provide
↪  your reasoning first before making a prediction. End your response by making a guess and saying 'The animal is:
↪  X' (e.g. The animal is: dog) at the end of your reasoning, where X is your guess. Do not provide any
↪  additional information. Your response should all fit in a single paragraph. If you are not provided any
↪  information, make a random guess.
```

---

**Input Prompt for Querier LLM Obtaining the Posterior**

```
You have not gathered any information yet. Please make a random guess. # If history is empty

Here is the information you have gathered. # If history is not empty
1. {q1} {q1(x)}
2. {q2} {q2(x)}
3. {q3} {q3(x)}
4. {q4} {q4(x)}
...

Given the information you have gathered, make an intermediate SINGLE prediction of what you think the animal is.
↪  First make your guess in the format 'The animal is: X' (e.g. The animal is: dog), where X is your guess, then
↪  provide your reasoning. Do not provide any additional information. Your response should all fit in a single
↪  paragraph. Make sure your prediction is one of the classes: {class_names}.
```

---

**Input Prompt for Querier LLM Suggesting Queries in Open Query Set Setting**

```
You have not gathered any information yet. Please make a random guess. # If history is empty

Here is the information you have gathered. # If history is not empty
1. {q1} {q1(x)}
2. {q2} {q2(x)}
3. {q3} {q3(x)}
4. {q4} {q4(x)}
...

Now suggest {n_queries_per_step} questions.\nReturn the question in this format:\n{{"questions": ["QUESTION_1",
↪  "QUESTION_2", "QUESTION_3"]}}
```

---

**System Instruction for Expert LLM for Non-binary Query Answers**

```
You are an expert on {label}. Based on the question provided, answer truthfully about the question. Do not
↪  directly tell the other player what you are thinking. Be as precise and as direct as possible, and answer in
↪  complete sentence. For example, if the question is "Does the animal have a tail?", you can answer "The animal
↪  has a tail." without saying yes or no. Do not say the name of the animal in your answer.
```

---

**System Instruction for Expert LLM for Binary Query Answers**

```
You are an expert on {label}. Based on the question provided, answer truthfully about the question. Do not
↪  directly tell the other player what you are thinking. Be as precise and as direct as possible, and answer with
↪  a single word. For example, if the question is "Does the animal have a tail?", you can answer "Yes." or "No.".
↪  Do not say the name of the animal in your answer. If you don't know the answer, make a guess. Do not answer
↪  anything other than "Yes." or "No.".
```

---

## H.2 MediQ

---

**System Instruction for Expert LLM Obtaining the Posterior**

```
You are a medical doctor specialized in {specialty}, trained to provide accurate, evidence-based responses to
↪  medical inquiries. Your goal is to answer questions with clarity, precision, and professionalism while
↪  ensuring your responses align with established medical guidelines. Answer concisely, accurately, and
↪  compassionately. Make a prediction and provide your reasoning as explanation. Respond in the following format:

{{"answer": "A/B/C/D", "explanation": "YOUR EXPLANATION HERE"}}
```

---

**Input Prompt for Expert LLM Obtaining the Posterior**

```
Answer the multiple choice based on the context.

Context: {context}

Question: {question}

Options:
    A - {option_a}
    B - {option_b}
    C - {option_c}
    D - {option_d}

Please select the most appropriate answer (A/B/C/D).
```

---

**System Prompt for Converting Facts into Queries**

```
Convert the medical fact into a question, in which the answer is the fact itself. The question should be specific
↪  and relevant to the patient's condition. Please do not ask any questions that are not related to the patient's
↪  medical history or condition. Suggest one question only. Return only your question and nothing else.

Medical fact: He has a non-productive cough for 4 months.
Question: What are some preliminary symptoms?

Medical fact: He complains of nausea and 1 episode of vomiting during the past day.
Question: Did the patient complain about nausea?
```

---

**System Prompt for Converting Facts into Queries**

```
Medical fact: {fact}
```

# I  Pseudocode for Implementation

---

**Algorithm 1** Constructing Prediction set functions for each length $\mathcal{C}_{\hat{\tau}_1} \ldots \mathcal{C}_{\hat{\tau}_L}$.

---

**Input:** Calibration set $\mathcal{D}_{\text{cal}}$ with $n_{\text{cal}}$ samples, Desired Coverage $1 - \alpha$, maximum iteration $L$, LLM-based predictor $f$.

1: **for** each length $k = 1 \ldots L$ **do**
2:
3:     *Step 1: Obtain conformal scores*
4:     **for** $(x_i, y_i) \in \mathcal{D}_{\text{cal}}$ **do**
5:         *Option 1.1: Uniform Sampling*
6:         Sample random history $q_{1:k} \sim Q_{1:k}$ based on Equation 13.
7:
8:         *Option 1.2: DP Sampling*
9:         Sample histories from direct prompting $q_{1:k} \sim \text{DP}(X, Y)$ based on Equation 14.
10:         Obtain scores based on LLM's output token logits for each class

$$s_i(x_i, y_i) = f(q_{1:k}(x_i))_{y_i}$$

11:     **end for**
12:
13:     *Step 2: Quantile Estimation*
14:     Estimate quantiles using the obtained scores:

$$\hat{\tau}_k = \widehat{\text{Quantile}} \left( \{s_i\}_{i=1}^{n_{\text{cal}}}; \frac{\lceil (n_{\text{cal}} + 1)(1 - \alpha) \rceil}{n_{\text{cal}}} \right)$$

15:
16:     *Step 3: Prediction Set Construction*
17:     Construct and define the prediction set function using the calculated quantile

$$\mathcal{C}_{\hat{\tau}_k}(q_{1:k}(x)) = \{y \in \mathcal{Y} \mid f(q_{1:k}(x)) > \hat{\tau}\}$$

18: **end for**
19: **return** Prediction set functions for each length $\mathcal{C}_{\hat{\tau}_1} \ldots \mathcal{C}_{\hat{\tau}_L}$.

---

**Algorithm 2** Information Pursuit with a closed query set $\mathcal{Q}_{\text{closed}}$.

---

**Input:** Observation or test sample $x^{\text{obs}}$, maximum iteration $L$, LLM-based predictor $f$, query set $\mathcal{Q} = \mathcal{Q}_{\text{closed}}$, estimation set $\mathcal{D}_{\text{est}}$
**Output:** Selected queries $q_{1:k}$ and predictions $\hat{y}_{1:k}$

1: Set iteration $k \leftarrow 0$
2: Initialize an empty history $\mathcal{S}_0 = \{\}$
3: **while** stopping criteria is met or $k < L$ **do**
4:     Predict $y_{k+1} = \text{argmax}_{y \in \mathcal{Y}} f(\mathcal{S}_k)_y$
5:     Estimate the entropy of $Y$ given each query $q$ using $\mathcal{D}_{\text{est}}$ select the most informative query

$$q_{k+1} = \underset{q \in \mathcal{Q}}{\text{argmin}}\, \hat{H}(Y \mid q(X), \mathcal{S}_k)$$

6:     Compute query answer $q_{k+1}(x^{\text{obs}})$ and update current history

$$\mathcal{S}_{k+1} = \mathcal{S}_k \cup \{q_{k+1}(x^{\text{obs}})\}$$

7:     $k \leftarrow k + 1$
8: **end while**

---

---
**Algorithm 3** Information Pursuit with a open query set $\mathcal{Q}_{\text{open}}$.

---
**Input:** Observation or test sample $x^{\text{obs}}$, maximum iteration $L$, LLM-based predictor $f$, an LLM LLM($\cdot$), $m$ queries to sample at each iteration, estimation set $\mathcal{D}_{\text{est}}$
**Output:** Selected queries $q_{1:k}$ and predictions $\hat{y}_{1:k}$
1: Set iteration $k \leftarrow 0$
2: Initialize an empty history $\mathcal{S}_0 = \{\}$
3: **while** stopping criteria is met or $k < L$ **do**
4:     Predict $\hat{y}_{k+1} = \text{argmax}_{y \in \mathcal{Y}} f(y \mid S_k)$
5:     Prompt language model for $m$ queries based on current history

$$\mathcal{Q} = \{q_j\}_{j=1}^m \leftarrow \text{LLM}(\mathcal{Y}, q_{1:k}(x^{\text{obs}}), m)$$

6:     Estimate the entropy of $Y$ given each query $q$ using $\mathcal{D}_{\text{est}}$ select the most informative query

$$q_{k+1} = \underset{q \in \mathcal{Q}}{\text{argmin}} \, \hat{H}(Y \mid q(X), \mathcal{S}_k)$$

7:     Compute query answer $q_{k+1}(x^{\text{obs}})$ and update current history

$$\mathcal{S}_{k+1} = \mathcal{S}_k \cup \{q_{k+1}(x^{\text{obs}}\}$$

8:     $k \leftarrow k + 1$
9: **end while**

---

---
**Algorithm 4** Conformal Information Pursuit with a closed query set $\mathcal{Q}_{\text{closed}}$.

---
**Input:** Observation or test sample $x^{\text{obs}}$, maximum iteration $L$, LLM-based predictor $f$, query set $\mathcal{Q} = \mathcal{Q}_{\text{closed}}$, Prediction set functions for each length $\mathcal{C}_{\hat{\tau}_1} \ldots \mathcal{C}_{\hat{\tau}_L}$, estimation set $\mathcal{D}_{\text{est}}$
**Output:** Selected queries $q_{1:k}$ and predictions $\hat{y}_{1:k}$
1: Set iteration $k \leftarrow 0$
2: Initialize an empty history $\mathcal{S}_0 = \{\}$
3: **while** stopping criteria is met or $k < L$ **do**
4:     Predict $\hat{y}_{k+1} = \text{argmax}_{y \in \mathcal{Y}} f(S_k)_y$
5:     Estimate the entropy of $Y$ given each query $q$ using $\mathcal{D}_{\text{est}}$ select the most informative query

$$q_{k+1} = \underset{q \in Q}{\min} \left\{ \lambda_\alpha + (1 - \alpha_N) \log \mathbb{E}_X [|\mathcal{C}_{\hat{\tau}_{k+1}}(q(X), \mathcal{S}_k)|] \right\}$$

6:     Compute query answer $q_{k+1}(x^{\text{obs}})$ and update current history

$$\mathcal{S}_{k+1} = \mathcal{S}_k \cup \{q_{k+1}(x^{\text{obs}})\}$$

7:     $k \leftarrow k + 1$
8: **end while**

---

**Algorithm 5** Information Pursuit with a open query set $\mathcal{Q}_{\text{open}}$.

---

**Input:** Observation or test sample $x^{\text{obs}}$, maximum iteration $L$, LLM-based predictor $f$, an LLM $\text{LLM}(\cdot)$, $m$ queries to sample at each iteration, Prediction set functions for each length $\mathcal{C}_{\hat{\tau}_1} \ldots \mathcal{C}_{\hat{\tau}_L}$, estimation set $\mathcal{D}_{\text{est}}$
**Output:** Selected queries $q_{1:k}$ and predictions $\hat{y}_{1:k}$
 1: Set iteration $k \leftarrow 0$
 2: Initialize an empty history $\mathcal{S}_0 = \{\}$
 3: **while** stopping criteria is met or $k < L$ **do**
 4:     Predict $\hat{y}_{k+1} = \text{argmax}_{y \in \mathcal{Y}} f(y \mid S_k)$
 5:     Prompt language model for $m$ queries based on current history

$$\mathcal{Q} \leftarrow \text{LLM}(\mathcal{Y}, q_{1:k}(x^{\text{obs}}), m)$$

 6:     Estimate the entropy of $Y$ given each query $q$ using $\mathcal{D}_{\text{est}}$ select the most informative query

$$q_{k+1} = \min_{q \in Q} \left\{ \lambda_\alpha + (1 - \alpha_N) \log \mathbb{E}_X [\|\mathcal{C}_{\hat{\tau}_{k+1}}(q(X), \mathcal{S}_k)\|] \right\}$$

 7:     Compute query answer $q_{k+1}(x^{\text{obs}})$ and update current history

$$\mathcal{S}_{k+1} = \mathcal{S}_k \cup \{q_{k+1}(x^{\text{obs}})\}$$

 8:     $k \leftarrow k + 1$
 9: **end while**

---

**Algorithm 6** Direct Prompting

---

**Input:** Observation or test sample $x^{\text{obs}}$, maximum iteration $L$, a Querier LLM QuerierLLM$(\cdot)$, a Expert LLM ExpertLLM$(\cdot)$, LLM-based predictor $f$
**Output:** Selected queries $q_{1:k}$ and predictions $\hat{y}_{1:k}$
 1: Initialize an empty history $\mathcal{S}_0 = \{\}$
 2: **for** $k = 0, \ldots, L$ **do**
 3:     Predict with $f$

$$\hat{y}_k = \underset{y \in \mathcal{Y}}{\text{argmax}} \, f(y \mid S_k)$$

 4:     Prompt Querier LLM for one single query given history

$$q_{k+1} = \text{QuerierLLM}(\mathcal{S}_k)$$

 5:     Obtain query answer and update history

$$q_{k+1}(x^{\text{obs}}) = \text{ExpertLLM}(q_{k+1}, x^{\text{obs}})$$
$$S_{k+1} = \mathcal{S}_k \cup \{q_{k+1}(x^{\text{obs}})\}$$

 6: **end for**

---

# J Additional Examples of Query-Answer Chains

In this section, we provide additional examples of query-answer chains obtained using C-IP. All the examples we provide in the following are correct predictions. Here we only show the iteration that the method stopped.

## J.1 20Q

Here we provide one example for each setting, which is either open or closed query set and either binary or free-text query answers.

---

**Example of 20Q with Closed Query Set and Binary Query Answers**

```
## System Instruction
You are an expert on animals. Your goal is to predict the animal given the information you have gathered. The
↪  animal must be one of the following: 'giraffe', 'zebra', 'elephant', 'killer whale', 'dalmatian', 'polar bear',
↪  'giant panda', 'hippopotamus', 'rhinoceros', 'lion', 'tiger', 'blue whale', 'walrus', 'grizzly bear','siamese
↪  cat', 'cow', 'german shepherd', 'gorilla', 'dolphin','moose'. Be as precise and as direct as possible. You may
↪  provide your reasoning first before making a prediction. End your response by making a guess and saying 'The
↪  animal is: X' (e.g. The animal is: dog) at the end of your reasoning, where X is your guess. Do not provide
↪  any additional information. Your response should all fit in a single paragraph. If you are not provided any
↪  information, make a random guess.

## Input Prompt
Here is the information you have gathered.
1. Does the animal live in the Arctic? No.
2. Does the animal have stripes? No.
3. Is the animal domestic? Yes.

Given the information you have gathered, make an intermediate SINGLE prediction of what you think the animal is.
↪  First make your guess in the format 'The ansmal is: X' (e.g. The animal is: dog), where X is your guess, then
↪  provide your reasoning. Do not provide any additional information. Your response should all fit in a single
↪  paragraph. Make sure your prediction is one of the classes: 'giraffe', 'zebra', 'elephant', 'killer whale',
↪  'dalmatian', 'polar bear', 'giant panda', 'hippopotamus', 'rhinoceros', 'lion', 'tiger', 'blue whale',
↪  'walrus', 'grizzly bear','siamese cat', 'cow', 'german shepherd', 'gorilla', 'dolphin','moose'.

## Response
The animal is: cow. Given the information that the animal lives in the Arctic is no, which eliminates polar bear,
↪  walrus, and grizzly bear, and that it is domestic, which eliminates most wild animals, leaving domesticated
↪  options such as a cow.
```

---

**Example of 20Q with Closed Query Set and Free-text Query Answers**

```
## System Instruction
You are an expert on animals. Your goal is to predict the animal given the information you have gathered. The
↪  animal must be one of the following: 'giraffe', 'zebra', 'elephant', 'killer whale', 'dalmatian', 'polar bear',
↪  'giant panda', 'hippopotamus', 'rhinoceros', 'lion', 'tiger', 'blue whale', 'walrus', 'grizzly bear','siamese
↪  cat', 'cow', 'german shepherd', 'gorilla', 'dolphin','moose'. Be as precise and as direct as possible. You may
↪  provide your reasoning first before making a prediction. End your response by making a guess and saying 'The
↪  animal is: X' (e.g. The animal is: dog) at the end of your reasoning, where X is your guess. Do not provide
↪  any additional information. Your response should all fit in a single paragraph. If you are not provided any
↪  information, make a random guess.

## Input Prompt
Here is the information you have gathered.
1. This animal primarily inhabits grasslands and savannas, rather than dense forests.

Given the information you have gathered, make an intermediate SINGLE prediction of what you think the animal is.
↪  First make your guess in the format 'The ansmal is: X' (e.g. The animal is: dog), where X is your guess, then
↪  provide your reasoning. Do not provide any additional information. Your response should all fit in a single
↪  paragraph. Make sure your prediction is one of the classes: 'giraffe', 'zebra', 'elephant', 'killer whale',
↪  'dalmatian', 'polar bear', 'giant panda', 'hippopotamus', 'rhinoceros', 'lion', 'tiger', 'blue whale',
↪  'walrus', 'grizzly bear','siamese cat', 'cow', 'german shepherd', 'gorilla', 'dolphin','moose'.

## Response
The animal is: zebra. This is because the information provided suggests a grassland or savanna habitat, which
↪  aligns with the typical habitats of zebras, as they are commonly found in grasslands and savannas.
```

## Example of 20Q with Open Query Set and Free-text Query Answers

## System Instruction
You are an expert on animals. Your goal is to predict the animal given the information you have gathered. The
↪ animal must be one of the following: 'giraffe', 'zebra', 'elephant', 'killer whale', 'dalmatian', 'polar bear',
↪ 'giant panda', 'hippopotamus', 'rhinoceros', 'lion', 'tiger', 'blue whale', 'walrus', 'grizzly bear','siamese
↪ cat', 'cow', 'german shepherd', 'gorilla', 'dolphin','moose'. Be as precise and as direct as possible. You may
↪ provide your reasoning first before making a prediction. End your response by making a guess and saying 'The
↪ animal is: X' (e.g. The animal is: dog) at the end of your reasoning, where X is your guess. Do not provide
↪ any additional information. Your response should all fit in a single paragraph. If you are not provided any
↪ information, make a random guess.

## Input Prompt
Here is the information you have gathered.
1. This animal primarily lives on land.
2. It is native to Europe and Asia, but was later introduced to the Americas and Africa.
3. This animal is a carnivore by natural instinct and primarily feeds on a diet rich in meat.

Given the information you have gathered, make an intermediate SINGLE prediction of what you think the animal is.
↪ First make your guess in the format 'The ansmal is: X' (e.g. The animal is: dog), where X is your guess, then
↪ provide your reasoning. Do not provide any additional information. Your response should all fit in a single
↪ paragraph. Make sure your prediction is one of the classes: 'giraffe', 'zebra', 'elephant', 'killer whale',
↪ 'dalmatian', 'polar bear', 'giant panda', 'hippopotamus', 'rhinoceros', 'lion', 'tiger', 'blue whale',
↪ 'walrus', 'grizzly bear','siamese cat', 'cow', 'german shepherd', 'gorilla', 'dolphin','moose'.

## Response
The animal is: dalmatian. The animal primarily lives on land, which matches with the fact that it is native to
↪ Europe and Asia, but was later introduced to the Americas and Africa, indicating a terrestrial habitat. This
↪ animal is a carnivore by natural instinct and primarily feeds on a diet rich in meat, which aligns with the
↪ characteristics of a dalmatian.

---

## Example of 20Q with Open Query Set and Binary Query Answers

## System Instruction
You are an expert on animals. Your goal is to predict the animal given the information you have gathered. The
↪ animal must be one of the following: 'giraffe', 'zebra', 'elephant', 'killer whale', 'dalmatian', 'polar bear',
↪ 'giant panda', 'hippopotamus', 'rhinoceros', 'lion', 'tiger', 'blue whale', 'walrus', 'grizzly bear','siamese
↪ cat', 'cow', 'german shepherd', 'gorilla', 'dolphin','moose'. Be as precise and as direct as possible. You may
↪ provide your reasoning first before making a prediction. End your response by making a guess and saying 'The
↪ animal is: X' (e.g. The animal is: dog) at the end of your reasoning, where X is your guess. Do not provide
↪ any additional information. Your response should all fit in a single paragraph. If you are not provided any
↪ information, make a random guess.

## Input Prompt
Here is the information you have gathered.
1. Does the animal primarily eat plants? No.
2. Does the animal primarily live on land? Yes.
3. Does the animal have four legs? Yes.
4. Does the animal have a large body size (over 400 kg)? Yes.
5. Does the animal live in a herd? No.
6. Is the animal known to be aggressive? Yes.
7. Does the animal have a long neck? No.
8. Does the animal have sharp claws? Yes.
9. Is the animal a carnivore? Yes.
10. Is the animal a mammal? Yes.

Given the information you have gathered, make an intermediate SINGLE prediction of what you think the animal is.
↪ First make your guess in the format 'The ansmal is: X' (e.g. The animal is: dog), where X is your guess, then
↪ provide your reasoning. Do not provide any additional information. Your response should all fit in a single
↪ paragraph. Make sure your prediction is one of the classes: 'giraffe', 'zebra', 'elephant', 'killer whale',
↪ 'dalmatian', 'polar bear', 'giant panda', 'hippopotamus', 'rhinoceros', 'lion', 'tiger', 'blue whale',
↪ 'walrus', 'grizzly bear','siamese cat', 'cow', 'german shepherd', 'gorilla', 'dolphin','moose'.

## Response
The animal is: lion. I based my prediction on the fact that the animal is a carnivore (9), lives on land (2), has
↪ a large body size (4), and is known to be aggressive (6). These characteristics are commonly associated with
↪ big cats, and the lion is one of the most aggressive and large land carnivorous mammals.

## J.2 MediQ

Here we provide one example for each specialty. Recall that, query answers (denoted as `Patient's Fact` in the prompt), is obtained in the order that C-IP obtains. When formatting as input, we ignore the query as that provides repetitive information as the query answer.

Some text that involves characters $\alpha, \beta, \mu$ are unable to render in our example, so we replace with with spelled out words, i.e. $\alpha \rightarrow$ "alpha".

---

**Example of Diagnosis in Internal Medicine**

```
## Input
You are a medical doctor answering real-world medical entrance exam questions. Based on your understanding of
↪   basic and clinical science, medical knowledge, and mechanisms underlying health, disease, patient care, and
↪   modes of therapy, answer the following multiple choice question. Select one correct answer from A to D. Base
↪   your answer on the current and standard practices referenced in medical guidelines. Repond in the following
↪   format:

{{"answer": "A/B/C/D", "explanation": "YOUR EXPLANATION HERE"}}

Answer the multiple choice based on the context.

Initial Info: A 54-year-old woman presents to the emergency department due to recent onset of a high fever,
↪   accompanied by severe headache and neck stiffness.

Conversation Log between doctor and patient:
Here is the information you have gathered.
Patient's Fact: The patient's current blood pressure reading is 112/71 mmHg.
Patient's Fact: She is currently on immunosuppressants after a kidney transplant 6 months ago. Her current
↪   medication regimen includes immunosuppressants.
Patient's Fact: The patient is 54 years old.
Patient's Fact: The patient has a history of rapidly progressive glomerulonephritis. She also underwent a kidney
↪   transplant 6 months ago and is currently on immunosuppressants.
Patient's Fact: The patient's current body temperature is 102.2°F (39°C).
Patient's Fact: The patient's neck pain is severe with neck flexion and reciprocal knee flexion. Facts 9 and 9 are
↪   the answer to this question.
Patient's Fact: The patient cannot answer the question about the typical triad of symptoms associated with
↪   meningitis.
Patient's Fact: Her respirations are 22/min.

Question: Which of the following characteristics is true of the most likely cause of this patient's disorder?
Options:
    A - Anaerobic spore forming rods
    B - Catalase-negative
    C - Coagulase-positive
    D - Tumbling motility in broth

Please select the most appropriate answer (A/B/C/D).

Medical conditions are complex, so you should seek to understand their situations across many features. First,
↪   consider which medical specialty is this patient's case; then, consider a list of necessary features a doctor
↪   would need to make the right medical judgment. Think step by step, reason about the patient information, the
↪   inquiry, and the options.

## Response
{"answer": "D", "explanation": "The patient's presentation of high fever, severe headache, neck stiffness, and
↪   neck pain with neck flexion and reciprocal knee flexion (positive Brudzinski's sign) suggests meningitis,
↪   which is an inflammation of the protective membranes covering the brain and spinal cord, known as the meninges.
↪   Given the patient's history of immunosuppression following a kidney transplant, she is at increased risk for
↪   opportunistic infections. The patient's inability to answer questions about the typical triad of symptoms
↪   associated with meningitis (headache, fever, and stiff neck) further supports this diagnosis.

The patient's age and immunosuppressed state make her susceptible to various pathogens, but the most likely cause
↪   of meningitis in this context is Listeria monocytogenes. Listeria monocytogenes is a gram-positive,
↪   catalase-negative bacterium that exhibits tumbling motility in broth. This characteristic is unique to
↪   Listeria species, making option D the correct answer.

Listeria monocytogenes is a common cause of meningitis in immunocompromised individuals, particularly those with a
↪   history of organ transplantation. It is also a cause of meningitis in pregnant women, the elderly, and
↪   individuals with certain underlying medical conditions. The other options do not accurately describe the
↪   typical characteristics of Listeria monocytogenes. Anaerobic spore-forming rods (option A) are more commonly
↪   associated with Clostridium species. Coagulase-positive (option C) is characteristic of Staphylococcus aureus,
↪   and while some anaerobic bacteria may exhibit catalase-negative (option B), this is not a specific
↪   characteristic of the most likely cause of this patient's disorder."}
```

## Input
You are a medical doctor answering real-world medical entrance exam questions. Based on your understanding of
↪ basic and clinical science, medical knowledge, and mechanisms underlying health, disease, patient care, and
↪ modes of therapy, answer the following multiple choice question. Select one correct answer from A to D. Base
↪ your answer on the current and standard practices referenced in medical guidelines. Repond in the following
↪ format:

{{"answer": "A/B/C/D", "explanation": "YOUR EXPLANATION HERE"}}

Answer the multiple choice based on the context.

Initial Info: An 8-year-old girl of Asian descent is brought to the physician because of fatigue.

Conversation Log between doctor and patient:
Here is the information you have gathered.
Patient's Fact: The patient's eye color and appearance is that she has pale conjunctivae.
Patient's Fact: The patient's mean corpuscular volume is 74 um3.
Patient's Fact: The chief complaint in this case is fatigue. She is not able to keep up with the rest of her
↪ classmates in gym class because she tires easily.
Patient's Fact: The patient tires easily during physical activities because she is not able to keep up with her
↪ classmates in gym class and is brought to the physician because of fatigue. This is also due to her pale
↪ conjunctivae and small, pale red blood cells.
Patient's Fact: She has a serum ferritin level of 77 ng/mL.
Patient's Fact: The patient's current Hemoglobin level is 11.0 g/dL.
Patient's Fact: She has normal hemoglobin electrophoresis. Hemoglobin electrophoresis is normal.

Question: Which of the following best describes the pathogenesis of the disease process in this patient?"
Options:
    A - Decreased production of beta-globin proteins
    B - Amino acid substitution in the beta-globin protein
    C - Cis deletion of alpha-globin genes
    D - Acquired inhibition of heme synthesis

Please select the most appropriate answer (A/B/C/D).

Medical conditions are complex, so you should seek to understand their situations across many features. First,
↪ consider which medical specialty is this patient's case; then, consider a list of necessary features a doctor
↪ would need to make the right medical judgment. Think step by step, reason about the patient information, the
↪ inquiry, and the options.

## Response
{"answer": "C", "explanation": "This patient's presentation of fatigue, pale conjunctivae, and small, pale red
↪ blood cells (microcytosis) is suggestive of anemia. The patient's mean corpuscular volume (MCV) of 74 um3
↪ indicates a microcytic anemia. The normal hemoglobin electrophoresis rules out hemoglobinopathies such as
↪ sickle cell disease or thalassemia major. The serum ferritin level of 77 ng/mL is elevated, which suggests
↪ iron overload. However, the patient's hemoglobin level is low, which is not consistent with iron overload. The
↪ most likely diagnosis is beta-thalassemia trait, which is a mild form of thalassemia. Beta-thalassemia trait
↪ is caused by a cis deletion of one of the beta-globin genes, leading to reduced production of the beta-globin
↪ protein. This results in a mild anemia, which is consistent with the patient's presentation. The patient's
↪ fatigue and pale conjunctivae are due to the anemia, and the small, pale red blood cells are consistent with
↪ microcytic anemia. The normal hemoglobin electrophoresis and elevated ferritin level support this diagnosis.
↪ Therefore, the correct answer is C - Cis deletion of alpha-globin genes is incorrect, as the patient has
↪ beta-thalassemia trait, not alpha-thalassemia. The correct answer is actually Cis deletion of beta-globin
↪ genes but the options provided do not have beta-globin, so the closest option is C."}
\end{lstlisting}

## Example of Diagnosis in Neurology

## Input
You are a medical doctor answering real-world medical entrance exam questions. Based on your understanding of
↪   basic and clinical science, medical knowledge, and mechanisms underlying health, disease, patient care, and
↪   modes of therapy, answer the following multiple choice question. Select one correct answer from A to D. Base
↪   your answer on the current and standard practices referenced in medical guidelines. Repond in the following
↪   format:

{{"answer": "A/B/C/D", "explanation": "YOUR EXPLANATION HERE"}}

Answer the multiple choice based on the context.

Initial Info: A 25-year-old man presents to the ER after a street fight.

Conversation Log between doctor and patient:
Here is the information you have gathered.
Patient's Fact: He has been rapidly decompensating since several hours after the initial hit. Since then, he has
↪   been rapidly decompensating.
Patient's Fact: The patient cannot answer the question about the exact cause of the altered mental status and head
↪   pain. However, the patient is experiencing head pain and altered mental status. The atomic factual statements
↪   that answer the question partially are: - He is grabbing his head in pain.- His mental status is altered.
Patient's Fact: The patient's right pupil is dilated. The information comes from fact #7.

Question: If this patient's symptoms are allowed to progress without intervention, which of the following would be
↪   expected to occur?
Options:
    A - Right sided hemiparesis
    B - Down and in gaze of the right pupil
    C - Right sided homonymous hemianopsia.
    D - Right eye with an inability to abduct

Please select the most appropriate answer (A/B/C/D).

Medical conditions are complex, so you should seek to understand their situations across many features. First,
↪   consider which medical specialty is this patient's case; then, consider a list of necessary features a doctor
↪   would need to make the right medical judgment. Think step by step, reason about the patient information, the
↪   inquiry, and the options.

## Response
{"answer": "A", "explanation": "Based on the patient's symptoms of rapidly decompensating altered mental status
↪   and head pain, along with a dilated pupil on the right side, the most likely diagnosis is a mass lesion in the
↪   brain, such as a hemorrhage or an ischemic stroke. The dilated pupil is a sign of third cranial nerve
↪   (oculomotor nerve) compression, which is often seen in patients with a space-occupying lesion in the brain.

If this patient's symptoms are allowed to progress without intervention, it is likely that the mass lesion will
↪   continue to increase in size and cause further brain damage. This could lead to a range of complications,
↪   including right-sided hemiparesis, which is weakness on the opposite side of the body from the lesion.

Right-sided hemiparesis is a common consequence of a left-sided brain lesion, as the motor tracts in the brain
↪   cross over to the opposite side of the body. The other options are not directly related to the expected
↪   outcome of a space-occupying lesion in the brain. Down and in gaze of the right pupil would be more consistent
↪   with a third cranial nerve palsy, but the patient's symptoms are not consistent with this diagnosis.
↪   Right-sided homonymous hemianopsia would be more consistent with a lesion in the visual pathway, and an
↪   inability to abduct the right eye would be more consistent with a sixth cranial nerve palsy, which is not
↪   suggested by the patient's symptoms."}

# K    Examples of Sampled Queries

Here we provide some brief examples of queries sampled in the 20Q open query set setting. Here the number of queries to sample $m = 3$.

---

**Example of Sampled Queries at Each Iteration**

```
[
    "Does the animal have four legs?",
    "Does the animal live on land?",
    "Is the animal a mammal?"
],
[
    "Does the animal have spots or stripes?",
    "Does the animal primarily live on land?",
    "Does the animal have a long neck?"
],
[
    "Does the animal primarily eat plants?",
    "Does the animal live in a group?",
    "Does the animal have a distinctive coat pattern?"
],
[
    "Does the animal have a long neck?",
    "Does the animal primarily eat bamboo?",
    "Does the animal give birth to a single cub at a time?"
],
[
    "Does the animal have a large body size?",
    "Does the animal primarily eat bamboo?",
    "Does the animal have a distinctive coat pattern?"
],
[
    "Does the animal primarily consume bamboo?",
    "Does the animal have retractable claws?",
    "Does the animal have a distinctive pattern of black and white stripes rather than spots?"
],
[
    "Does the animal primarily feed on plants?",
    "Does the animal have a herbivorous diet?",
    "Is the animal native to Asia?"
],
[
    "Does the animal primarily feed on bamboo?",
    "Is the animal a carnivore?",
    "Does the animal have a large body size compared to other land animals?"
],
[
    "Does the animal primarily walk on all fours?",
    "Does the animal have a long gestation period?",
    "Is the animal a member of the Ursidae family?"
],
[
    "Does the animal have a long snout?",
    "Is the animal native to Africa?",
    "Does the animal have a large body size compared to its neck?"
],
[
    "Does the animal primarily live on land?",
    "Is the animal a carnivore?",
    "Does the animal have a large body size compared to other animals on the list?"
]
```

