# OpenReview forum: "Conformal Information Pursuit for Interactively Guiding Large Language Models"
_NeurIPS.cc/2025/Conference — NeurIPS 2025 poster_

### Official Review · Reviewer_zf88 · 2025-06-23

**Clarity:** 3
**Significance:** 3
**Originality:** 3
**Rating:** 4
**Confidence:** 4

**Summary:**

C-IP provides a mathematically rigorous solution to uncertainty quantification in sequential querying, using conformal prediction sets to address fundamental LLM calibration issues. While offering strong theoretical guarantees and demonstrated performance improvements, the approach has limited scope beyond sequential questioning scenarios and does not directly enhance reasoning quality. The contribution is significant within its domain, with clear potential for broader application and integration with complementary reasoning frameworks.

**Questions:**

Good work.  A couple of questions...

1. Your method's effectiveness appears heavily dependent on the quality and comprehensiveness of the available query set Q. In real-world scenarios where optimal queries may not be pre-known or where the query space is naturally limited, how robust is C-IP to suboptimal query sets? Could you provide guidance on when practitioners should expect the method to fail or underperform?

2. While C-IP effectively addresses uncertainty quantification, it doesn't directly improve the quality of reasoning or handle LLM hallucinations. Given the demonstrated complementarity with approaches like EVINCE that focus on reasoning quality, have you considered how C-IP could be integrated into broader multi-LLM reasoning frameworks? What would be the technical challenges and potential benefits of such integration?

**Ethical Concerns:**

["NO or VERY MINOR ethics concerns only"]

**Final Justification:**

The paper presents a theoretically sound and empirically validated method for interactive LLM querying. However, its contributions are incremental and the scope narrow, leading to my weak accept rating. The proposed method acts as a bandaid to improve LLM answer quality—greedy in nature, with performance heavily dependent on data and query specifics.

I acknowledge the rebuttal from the authors to be reasonable, but the weak accept rating already reflects my respect to the work with the following weaknesses:

Bottom Line Weaknesses

Narrow focus - Only addresses calibration, ignores other critical LLM problems
Limited novelty - Core information-theoretic approach may not be sufficiently novel
Practical limitations - Requires good query sets, unclear how to scale (RAG may not guarantee high quality data)

**Limitations:**

Yes

**Quality:**

3

**Strengths And Weaknesses:**

A. Strengths of this work:

A.1. Rigorous Theoretical Foundation: The work is grounded in conformal prediction theory, providing mathematically principled uncertainty quantification with formal coverage guarantees. Notably, the approach is distribution-free and does not require calibrated probability estimates, addressing a fundamental limitation of traditional entropy-based methods.

A.2 Demonstrated Practical Effectiveness: C-IP shows consistent performance improvements across multiple domains (20 Questions, medical QA), with clear mathematical stopping criteria and computational efficiency for discrete classification problems. The method provides robust uncertainty estimates even when LLM probabilities are miscalibrated.

B. Weaknesses of this work:

B.1 Limited Scope and Applicability:

Narrow focus on sequential query scenarios limits broader applicability
Primarily designed for discrete classification problems
Heavy dependence on query set quality and availability
No mechanisms for improving reasoning quality or argument coherence

B.2 Fundamental Methodological Gaps:

Does not address core LLM limitations (bias, hallucination, error propagation)
Static approach lacks adaptive mechanisms for evolving problem complexity
Limited to uncertainty quantification without enhancing underlying reasoning processes

Overall Assessment: C-IP makes a significant contribution to uncertainty quantification in sequential querying through rigorous mathematical foundations. However, its impact could be substantially enhanced through integration with complementary reasoning frameworks like EVINCE [1], which addresses the broader challenges of LLM reliability and reasoning quality that C-IP does not tackle.

[1] Optimizing Multi-LLM Dialogues Through Information-Theoretic Control, E. Chang, August 2024.
https://www.researchgate.net/publication/383425368_Optimizing_Multi-LLM_Dialogues_Through_Information-Theoretic_Control

---

> ### Author Rebuttal · Authors · 2025-07-30
>
> Thank you for the positive review. We are glad to hear that you find our work to have "Rigorous Theoretical Foundation" and "Demonstrated Practical Effectiveness." Please see below for our response.
>
> **Limited Scope and Applicability**
>
> We respectfully disagree that our paper has limited scope and applicability. To the best of our knowledge, the setting of interactions between multiple LLMs by maximizing information gain is a relatively unexplored direction. We have also demonstrated via interactive medical question answering setting that it can be extended to real-world tasks. Our thorough experiments, we argue, provide a strong justification for the applicability and potentiality of our work.
>
> Last but not least, our method does not explicitly consider reasoning. We neither require or leverage reasoning LLMs. Therefore, we believe that ``improving reasoning quality or argument coherence" is a completely orthogonal problem to what is considered in our paper.
>
>
>
>  **Fundamental Methodological Gaps**
>
> We address the major "core LLM limitations" relevant to our problem.  For our problem, the core issue with LLMs, as stated in our introduction, is the fact that LLMs probabilities are miscalibrated. This also serves as our motivation to develop our proposed method C-IP. While bias, hallucination, error propagation are all important research directions, they are not as important for our specific problem. For instance, when interacting in a 20 questions setting, LLMs must simply choose among 20 different possibilities, and so issues of bias or error propagation are not the most salient ones. While improving performance on these issues, We believe that there is higher payoff to remedy calibration problems and that's why we focused on that task here.
>
> We disagree that our work "lacks adaptive mechanisms." Our method C-IP is adaptive by design: C-IP is a sequential algorithm, and selects/answers queries based on the provided context and information obtained adaptively.
>
> Finally, we are unclear what the reviewer is referring to as "reasoning process," as our work involves no LLM reasoning (In the sense of modern reasoning LLMs such as o1, o3, or DeepSeek).
> If this is meant informally, then we would like to clarify that our method does in fact improve reasoning (compared to standard baselines such as information pursuit or chain of thought), by appropriately prioritizing the questions to ask.
>
> **Questions and Comments about Query set, such as "Heavy dependence on query set quality" and "optimal queries may not be pre-known"**
>
> Certainly, there is some mild dependence on the query set, but this is reasonable and easily satisfied at least approximately in practice.  Basically, in our experiments we found that our method works well with a variety of query sets.
> From the definition of our problem (see equation (1)), we assume that our query set is *sufficient*: which means that the set of all queries fully describes the given task $Y$. In our work, we have explored both settings where query set is known ahead of time (closed query set) and is unknown (open query set). In both settings, we find that C-IP is not impacted by the fact that the query set may not be known ahead of time.
>
> In our experiments, we have demonstrated that our method works well in both closed query set setting where the user predefined a fixed query set and open query set setting where the queries are prompted from LLMs.
>
> Moreover, as a general practical strategy to approximately satisfy sufficiency,
> one may resort to pre-processing methods such as retrieval augmented generation with knowledge bases or prompting techniques to acquire a sufficient query set. While this direction has been explored in previous versions of IP [1], the task of generating good questions/queries from LLMs remains a worthy research direction.
>
> [1] Aditya Chattopadhyay, Kwan Ho Ryan Chan, and Rene Vidal. "Bootstrapping variational information pursuit with large language and vision models for interpretable image classification." In The Twelfth International Conference on Learning Representations. 2024.
>
> **Complementary Frameworks such as EVINCE**
>
> Thank you for the suggestion. We were unaware of this work and will add this to our related work section; we agree that the work is complementary and could be incorporated in our framework. We will reserve this as a future direction.
>
> In our work, we have presented the simplest setting for interacting with LLMs, selecting queries in the order of information gain. We agree that this setting can potentially be extended from simple to complex multi-LLM settings, such as reaching consensus and forming arguments and debating. As an example, it might require us to modify the query-answering mechanism to allow for multi "expert" LLMs, or modify the objective to allow for estimating mutual information among multiple LLMs. We argue that the method and framework we have presented in this framework, because of its simplicity, allows a multitude of modifications that can adapt to many current, exciting LLM directions, including the EVINCE framework.

---

> > ### Comment · Reviewer_zf88 · 2025-08-05
> >
> > The paper presents a theoretically sound and empirically validated method for interactive LLM querying. However, its contributions are incremental and the scope narrow, leading to my weak accept rating. The proposed method acts as a bandaid to improve LLM answer quality—greedy in nature, with performance heavily dependent on data and query specifics. More principled approaches exist for tackling this class of problems, though I cannot elaborate without risking reviewer anonymity. In summary, the rating reflects respect for the effort, despite reservations about the broader impact.

---

### Official Review · Reviewer_QsyY · 2025-06-29

**Clarity:** 3
**Significance:** 3
**Originality:** 3
**Rating:** 5
**Confidence:** 3

**Summary:**

In the paper an Information Pursuit strategy is applied to settings where LLMs can interactively query the user to make a prediction, e.g. in a medical diagnosis setting. Such strategies typically maximize the information gain (or equivalently the conditional entropy) of each query. While LLMs output probabilities over their outcome, these probabilities are known to be miscalibrated and therefore do not serve as a reliable measure of uncertainty need to estimates the conditional entropy. The paper therefore proposes to leverage the upper bound on the entropy base on the cardinality of prediction sets (see Correira et al.), in order to circumvent the miscalibration issue.

**Questions:**

* For the results in Figure 2: How is "uncertainty" defined and measured?
---

**An explanation regarding the following two questions would likely make me increase my score:**

---
* Since the LLM predictions are miscalibrated, the sampled of histories from LLM simulations should also be biased and therefore the predictions sets, too, might be biased. Is it some kind of "averaging out" of biases that makes the construction work well in the end nevertheless?
---
* Section 5.2 says:
> The performance of C-IP and the baselines for the three specialities is shown in Figure 4. C-IP consistently obtains higher test accuracy at each iteration than IP for IM and P while achieving comparable performance for N.

In Figure 4 on the **IM dataset**, to me it looks like that C-IP has lower accuracy for 1,2,5,6 and 7 iterations and higher accuracy for 3,4, 9 and 10 iterations. On the **P dataset**, it looks like C-IP performs better for 2,3,4,7,9 and 10 iterations and worse for 1,5,6 and 8 iterations. Am I misreading Figure 4? Otherwise, I would suggest weakening the statement "consistently obtains higher test accuracy" to "achieves comparable performance".

**Ethical Concerns:**

["NO or VERY MINOR ethics concerns only"]

**Final Justification:**

My concerns were addressed with the additional clarifications and I therefore updated my score to recommend acceptance.

**Limitations:**

yes

**Paper Formatting Concerns:**

no formatting concerns

**Quality:**

3

**Strengths And Weaknesses:**

**Strengths:**
* the approach is theoretically sound as it is based on a mathematical relation between the entropy and the cardinality of prediction sets.
* reducing the number of questions in such interactive LLM-settings is an important research aim and the suggested approach is as far as I know an original combination of the Information Pursuit, Prediction sets and LLMs.
* clear motivation and easy-to-follow introduction of the background material on Information pursuit and conformal prediction.

**Weaknesses:**
*  it is not fully clear to me if there are performance gains in the application setting in Section 5 or not

**Minor comment:**
* The legend in Figure 4 is very small and difficult to read.
* It is fairly uncommon to have the Related Work Section in the Appendix only. Usually explaining how the work builds upon the existing literature is an integral part of a paper.

---

> ### Author Rebuttal · Authors · 2025-07-30
>
> Thank you for your positive. We appreciate the clear instructions on how to improve our scores. We are glad you find our paper "theoretically sound" with "clear motivation."  Also, thank you for acknowledging that our direction of "interactive LLM-settings is an important research." Please kindly see below for our response.
>
> **Definition and Measurement of Uncertainty in Figure 2**
>
> Thank you for the clarification question. Figure 2 is intended to serve as a motivating figure that abstracts from Figure 3. The results shown here is exactly that from 20Q with Llama-3.1-8B with a closed query set. The dashed curve corresponds to the measurement of conditional entropy $H(Y \mid q(X), H\_k)$, directly estimated from probabilities obtained from LLM logits. The solid curves correspond to the measurement of expected size of the prediction set $E\_{X}[C(q(X)) \mid H_k]$.
>
> **"Bias of prediction sets due to bias in sampled histories from LLMs"**
>
> We thank the reviewer for the clarification question. The issue of bias is exactly the reason why conformal prediction is needed, and the process of finding the threshold $\tau$ based on the conformal guarantee is the exact step in accounting for this bias. To be more explicit, let's consider the construction of the prediction set $C_\tau$: Suppose we fix $\alpha = 0.05$. From the calibration set $D\_{cal} = \{(x\_i, y\_i)\}\_{i=1}^{N}$ and their sampling of histories, we can observe the probability of the correct label $P(Y = y\_i \mid H\_k)$. If the probabilities are biased towards something small, say below 0.5, then the threshold $\tau$ we obtained will also be smaller than 0.5. In fact, $\tau$ is the threshold where $100(1-\alpha)$\% of the correct samples are above the threshold. In other words, the biases are considered and adjusted in the process of conformal prediction itself. Therefore, assuming test samples and calibration samples are exchangeable, we are effectively accounting for this bias during inference.
>
> **Performance Gains and Question regarding Figure 4**
>
> Thank you for your comment. We agree that, relative to Section 4 and experiments with 20Q, the performance gains shown in Section 5 with Interactive Medical Question Answering is minor. For Figure 4, we agree that the statement is slightly over-claimed and will change it to "achieves comparable performance".
>
> However, we would also like to emphasize that the purpose is to demonstrate the potential and broad applicability of C-IP to real-world settings. It's worth noting that we do observe performance gains of C-IP over sequential baseline (enumerate).
>
>
> **Minor Comments about Formatting and Related Work**
>
> Thank you for the suggestions. We will increase the size of the legend to make it more readable. As for the Related Work section, at the time of writing we believed that the introduction and build-up of the work has sufficiently mentioned the nearest neighbors of our work. However, since it is not a complete full-fetched Related Work section, we decided to add a full section in the appendix for coherence and clarification.

---

> > ### Comment · Reviewer_QsyY · 2025-08-05
> >
> > Thank you for the reply. I am still not sure I understand the part regarding the "Bias of prediction sets due to bias in sampled histories from LLMs":
> > * Where do you get the calibration set from in general?
> > * Where in the paper do I find information what the calibration sets were in your experiments?

---

> ### Author Response · Authors · 2025-08-06
>
> Thank you for the clarification question.
>
> Generally in conformal prediction, the calibration set contains samples from the distribution for which we are aiming to achieve coverage. For instance in the simpler setting of predicting $Y$ from $X$ that are drawn from a given distribution where the coverage guarantee is $P_{X, Y}(Y \in C(X)) \ge 1 - \alpha$, the calibration set is $D_{cal} = \\{(x_i, y_i)\\}_{i=1}^{N}$, where each datapoint is drawn from the same distribution. For a fuller explanation, please refer to [1].
>
> In our case, since we are trying to satisfy the coverage guarantee (Equation 11), the calibration set contains samples of the query-answer chains. We proposed two ways of obtaining samples and the calibration set, which depends the setting of closed/open query set: (1) calibration samples in the closed query set setting are obtained from a uniform parameterization described in L204-209, and (2) calibration samples in the open query set setting are obtained by LLM simulations described in L210-214 as well as Appendix I, Algorithm 6. As to the number of samples, we use 100 samples for 20Q and 200 samples for MediQ, both described in L251, L345, and Appendix G.7. We will update the Appendix section G.7 with a more complete description to make this entire procedure clear.
>
> Therefore, to add to our original comment about "Bias of prediction sets due to bias in sampled histories from LLMs", the bias of in sampled histories from LLMs is based on the empirical distribution of samples we obtain, in which the conformal prediction component of our method directly addresses by satisfying our desired coverage (Equation 11).
>
> [1] Angelopoulos, A. N., \& Bates, S. (2021). A gentle introduction to conformal prediction and distribution-free uncertainty quantification. arXiv preprint arXiv:2107.07511.

---

> > ### Comment · Reviewer_QsyY · 2025-08-07
> >
> > Thank you for the additional clarification. I think I understand it now. I will update my score accordingly.

---

### Official Review · Reviewer_spQy · 2025-07-02

**Clarity:** 3
**Significance:** 2
**Originality:** 2
**Rating:** 4
**Confidence:** 3

**Summary:**

This paper introduces conformal information pursuit, which extends the information pursuit framework for guiding LLMs in sequential question selection. Specifically, C-IP uses conformal prediction sets to estimate uncertainty in a distribution-free manner for subsequent query selection. The method is evaluated on the 20 Questions and MediQ datasets.

**Questions:**

See above.

**Ethical Concerns:**

["NO or VERY MINOR ethics concerns only"]

**Final Justification:**

The authors partially addressed my concerns about comparisons. So, I have updated my score to 4.

**Limitations:**

yes.

**Quality:**

2

**Strengths And Weaknesses:**

Strengths

- The proposed framework is easy to implement.

- The study is well-motivated by the open-ended nature of queries and answers in conversational settings.

Weaknesses

1. Lack of Theoretical Contribution

- The study relies heavily on theoretical analysis from prior work [18, 64], with limited novel theoretical development.

2. Insufficient Theoretical Justification for History Sampling and Multi-Round Settings

- The construction of prediction sets across dialogue histories involves marginalization steps that lack theoretical grounding.

- Additionally, the paper proposes two potential methods for history sampling, but neither is supported by formal theoretical analysis.

3. Limited Baselines and Marginal Improvements

- There is no comparison to learned query policies, calibration-based methods, or reinforcement learning-based strategies.

- The performance improvements appear modest and primarily limited to baseline comparisons, with gains mostly in early stages.

4. Why Not Calibrate the LLM Instead?

- The paper assumes that LLM outputs are too miscalibrated for entropy-based selection, but makes no effort to calibrate the LLM’s own confidence scores.

5. Similarity to Prior Work

- The proposed method appears highly similar to [64].

---

> ### Author Rebuttal · Authors · 2025-07-30
>
> Thank you for your review and questions. We are glad to hear that you find our "well-motivated by the open-ended nature of queries and answers in conversational settings." Please kindly see below for our responses.
>
> **"Highly Similar to Conformal Language Modelling [64]"**
>
> We strongly disagree that our work is similar to Conformal Language Modeling (CLM) [64]. They develop a method for conformal language modelling, which requires repeated (iid, non-interactive) sampling from an LLM until a certain correctness criterion is met, using conformal risk control.
> Our work focuses on interacting with LLMs via information pursuit by
> seeking to ask informative questions. We start from Information Pursuit (IP), entropy maximization, and leverage conformal prediction to get a feasible method. Notably, CLM uses conformal risk control, while we connect to standard conformal prediction as one way of estimating uncertainty. Thus, while LLMs and conformal prediction are used in both our work and CLM, our problem setting, formulation, and experiments cannot be farther apart from those of CLM.
>
>
> **The lack of theoretical justification**
>
> The central contribution of our work is methodological, not theoretical. Specifically, we propose an efficient method to interactively guide an LLM by choosing queries based on maximizing information gain and use of conformal prediction to improve query selection with IP. Thus, we politely but firmly disagree with the reviewer that we would need novel theoretical developments (given the additional significant methodological innovation). Our work is theoretically inspired by the work [18]. Regarding the marginalization step and sampling methods, as we already stated and justified in the paper, they well in practice and in our experiments. We argue that a complete theoretical justification is beyond the scope of our work; because this goes well beyond the current theoretical frontier in conformal prediction and may require significant and completely novel theoretical approaches.
>
> Following current standards in the field (as can be checked in papers in recent conferences), having theoretical contributions in a methodological paper is not a requirement to acceptance at NeurIPS.
>
> **Lack of Baselines and Marginal Improvements**
>
> We have already provided thorough comparisons to state-of-the-art methods, including information pursuit and Uncertainty-of-Thought, an entropy-based chain-of-thought method applicable to binary queries. We have compared to all methods that we know of to have high performance on the problems of iterative information seeking; following best practices in the area. We found considerable performance gains (say a success rate increase from 0.5 to 0.8).
>
> Regarding other comparisons, we do not think that there is a clear RL baseline that is directly comparable; as one would have to define an appropriate reward, environment, etc.. We are not aware of appropriate existing ``learned query policies" that one could use for information seeking. Can you clarify which related works exactly you mean should be compared?
>
> Furthermore, our work focuses on the strategy of IP that yields short query-answer chains; we argue performance of earlier iterations demonstrate ability to choose informative queries, and performance of later iterations demonstrate ability to converge to the full posterior with a subset of queries (i.e. $KL(P(y | x) \mid P(y \mid q\_{1:k}(x)))$). Overall, our work focuses on interactivity with information gain, a dimension of LLM research that remains largely unexplored.
>
>
> **"Why not just calibrate the LLM instead?"**
>
> Thank you for the additional baseline suggestion. We agree that there is the possibility that the alternative method of "first calibrating LLM probabilities then estimate uncertainty using entropy" might find success.
> To test this hypothesis, we performed additional experiments by first applying temperature scaling (TS) and Platt scaling (PS) to calibrate LLM probabilities, then inference with standard IP (not the conformal version). Note that TS (for a fixed temperature) does not depend on calibration data, whereas PS does depend on calibration data. We evaluate 20Q in two of the open query set settings as before, and average each accuracy curve over 5 random seeds.
>
> *Open query set, Binary query answers*
> | Iteration             |        0    |        1 | 2    | 3                | 4    | 5    | 6    | 7    | 8    | 9    | 10   | 11   | 12   | 13   | 14   | 15   | 16   | 17   | 18   | 19   | 20   |
> |-----------------------|-------------|----------|------|------------------|------|------|------|------|------|------|------|------|------|------|------|------|------|------|------|------|------|
> | IP w/ TS ($T=0.5$)    | 0.05        | 0.12     | 0.2  | 0.2              | 0.26 | 0.29 | 0.29 | 0.29 | 0.31 | 0.33 | 0.34 | 0.34 | 0.36 | 0.36 | 0.39 | 0.41 | 0.41 | 0.43 | 0.43 | 0.44 | 0.45 |
> | IP w/ TS ($T=0.75$)   | 0.05        | 0.07     | 0.11 |             0.11 | 0.14 | 0.17 | 0.21 | 0.23 | 0.26 | 0.27 | 0.29 | 0.29 | 0.31 | 0.32 | 0.32 | 0.32 | 0.32 | 0.34 | 0.36 | 0.37 | 0.37 |
> | IP w/ TS ($T=1.25$)   | 0.05        | 0.1      | 0.18 | 0.26             | 0.27 | 0.3  | 0.35 | 0.38 | 0.41 | 0.42 | 0.42 | 0.42 | 0.44 | 0.46 | 0.46 | 0.47 | 0.48 | 0.48 | 0.48 | 0.49 | 0.49 |
> | IP w/ Platt Scaling   |        0.05 | 0.17     | 0.23 | 0.27             | 0.3  | 0.33 | 0.4  | 0.42 | 0.43 | 0.45 | 0.46 | 0.46 | 0.47 | 0.47 | 0.48 | 0.51 | 0.51 | 0.51 | 0.51 | 0.52 | 0.53 |
> | C-IP ($\alpha = 0.1$) |        0.05 | 0.13     | 0.19 | 0.23             | 0.31 | 0.35 | 0.41 | 0.43 | 0.44 | 0.45 | 0.46 | 0.46 | 0.48 | 0.51 | 0.52 | 0.54 | 0.57 | 0.58 | 0.58 | 0.59 | 0.59 |
>
> *Open query set, Free-text query answers*
> | Iteration              |        0    |        1 | 2    | 3    | 4    | 5    | 6    | 7    | 8    | 9    | 10   | 11   | 12   | 13   | 14   | 15   | 16   | 17   | 18   | 19   | 20   |
> |------------------------|-------------|----------|------|------|------|------|------|------|------|------|------|------|------|------|------|------|------|------|------|------|------|
> | IP w/ TS ($T=0.5$)     | 0.05        | 0.25     | 0.44 | 0.49 | 0.57 | 0.61 | 0.69 | 0.71 | 0.74 | 0.81 | 0.81 | 0.84 | 0.84 | 0.84 | 0.84 | 0.85 | 0.85 | 0.85 | 0.85 | 0.85 | 0.86 |
> | IP w/ TS ($T=0.75$)    | 0.05        | 0.28     | 0.44 | 0.56 | 0.62 | 0.62 | 0.64 | 0.69 | 0.69 | 0.71 | 0.75 | 0.77 | 0.79 | 0.79 | 0.79 | 0.79 | 0.79 | 0.79 | 0.79 | 0.79 | 0.79 |
> | IP w/ TS ($T=1.25$)    | 0.05        | 0.34     | 0.47 | 0.57 | 0.61 | 0.66 | 0.69 | 0.72 | 0.78 | 0.81 | 0.81 | 0.84 | 0.84 | 0.84 | 0.84 | 0.84 | 0.85 | 0.85 | 0.86 | 0.86 | 0.86 |
> | IP w/ Platt Scaling    |        0.05 | 0.3      | 0.43 | 0.52 | 0.61 | 0.64 | 0.74 | 0.75 | 0.77 | 0.81 | 0.82 | 0.84 | 0.84 | 0.88 | 0.89 | 0.89 | 0.9  | 0.91 | 0.91 | 0.91 | 0.91 |
> | C-IP ($\alpha = 0.15$) |        0.05 | 0.35     | 0.53 | 0.6  | 0.67 | 0.72 | 0.75 | 0.8  | 0.81 | 0.83 | 0.86 | 0.88 | 0.88 | 0.89 | 0.9  | 0.91 | 0.91 | 0.91 | 0.91 | 0.91 | 0.91 |
>
> For TS, logits obtained from LLMs are divided by a temperature scalar $T \in \\{0.5, 0.75, 1.25\\}$ before applying softmax. We observe that C-IP outperforms IP with TS in all settings.
>
> For standard calibration methods that require calibration data (e.g. PS), we note that we cannot directly "plug-in" and apply; similar to C-IP with marginal guarantee over only samples, they still require us to obtain samples with the exact history $q_{1:k}(x^{\text{obs}})$ during inference (See L190-195 for our original, full argument). Hence, to run Platt scaling with calibration data, we also need to do random sampling as for our method. Here, we will run IP with Platt scaling using the same calibration data as our experiments with C-IP. In our additional experiments, we found that IP with PS performs worse in the earlier iterations but comparably in later iterations for the free-text query answers setting. Moreover, IP with PS performs worse in all iterations for the binary query answers setting.
>
> Overall, our additional baseline results find that C-IP outperforms IP with calibration methods of TS and PS. We argue that in the open query set settings, where the distribution is more complex, conformal prediction is a better method for good estimations of uncertainty, showing that calibration alone is not sufficient and our formulation is necessary.

---

> ### Author Response · Authors · 2025-08-04
>
> Thank you again for the insightful reviews. If there are any unresolved concerns and questions, feel free to ask any additional questions to which we may answer and provide additional clarification. If you feel our rebuttal has sufficiently addressed your concerns, we would appreciate it if you could kindly reconsider your evaluation.
>
> Sincerely,
>
> Authors

---

### Official Review · Reviewer_HBRz · 2025-07-04

**Clarity:** 3
**Significance:** 3
**Originality:** 3
**Rating:** 5
**Confidence:** 4

**Summary:**

This paper looks at interactive multi-turn dialogues where an LLM is able to ask questions and obtain information before giving a final prediction (e.g., as in the 20 Questions game). The strategy the authors pursue is motivated by "information pursuit" (or choosing to ask queries that maximize information gain at each step), but via the use of conformal prediction sets instead of measures of conditional entropy that rely on estimated probabilities from the LLM (which are unreliable). Specifically, the approach is to minimize prediction set size at each iteration --- which the authors connect to an upper bound on the standard entropy. Empirically, the authors show good performance on 20Q + MediQ at getting good predictive accuracy within a small number of iterations.

**Questions:**

- I'm a bit confused by L169, where the claim is made the the uncertainty of Y given data X is upper bounded by the expected size of the prediction set. I think the more precise claim is + some additional constants that depend on $\alpha$ and $|\mathcal{Y}|$ (which, looks like it can be quite large, unless $\alpha$ is near $0$ --- for instance, for $\alpha$ near 1 this bound is vacuous, as it bounds the conditional entropy by the entropy of a uniform distribution over $\mathcal{Y}$). It's also weird to me to related the conditional entropy to the expected set size, marginal over $X$ --- this is then a bound that is no longer dependent on the input example?

- I'm also a bit unclear on how $E_X[ \mathcal{C}(q(X)) \vert \vert \mathcal{H}]$ is computed. Are the same set of query trajectories applied to the calibration set for all $X$? In the open query set, what happens when $q$ does not apply to some $X_i \in \mathcal{D}_\mathrm{cal}$?

- The calibration set is used quite extensively for the CP algorithm. Have you tried also using it for the baselines, e.g., to calibrate the confidence of the LLMs by histogram binning or platt scaling?

**Ethical Concerns:**

["NO or VERY MINOR ethics concerns only"]

**Final Justification:**

I think that some of the theory is a bit "fluff" --- I'm not quite convinced that the exercise to connect information pursuit to conformal prediction is necessary, I still think it seems likely that a baseline around discretizing uncertainty estimates and calibrating them on a calibration set would yield similar gains, though I'm not exactly sure what it would be. That said, this is definitely and interesting method, and does achieve that goal (better sequential information pursuit) with encouraging empirical results.

**Limitations:**

Yes

**Quality:**

3

**Strengths And Weaknesses:**

The paper is well-written, and the motivation (reducing uncertainty set size) is well-motivated. The experiments also indicate that the proposed method is effective. However, I also have a number of questions (see below), and at a high level, while I buy that a measure that is less susceptible to estimation error like conditional entropy is, I am not fully convinced of the need for _conformal_ prediction and calibration. Specifically, the primary advantage of conformal prediction---the formal coverage guarantee---is not a central feature of the algorithm's goal, which is to minimize uncertainty for query selection, not to produce prediction sets with valid coverage (and even so, the benefits of the marginal conformal guarantees are somewhat dubious here --- which is made even more marginal here due to tractability issues). For me, this raises a bit of a question of whether the complexity of the conformal framework is justified over simpler, alternative uses of the calibration data (which does not seem to be used for the baselines).

More concretely, the proposed C-IP method uses the average size of a conformal prediction set as a proxy for uncertainty. The size of the conformal sets also depend on the LLM probabilities, but in a more discretized way, i.e., the count of how many outputs have confidence > $\tau$. $\alpha$ doesn't seem to be used in any important way here, so searching over $\alpha$ hyper-parameters I believe could also be viewed as just measuring the average number of labels per X that are above some confidence threshold. I would be curious to compare this to other, simple baselines that use metrics like this --- like the average top-label confidence, the average difference between the top label's confidence and the second-to-top label's confidence, or if the confidence score is noisy, the average accuracy of the top-label (so, ranking queries by their conditional accuracy).

---

> ### Author Rebuttal · Authors · 2025-07-30
>
> Thank you for your review and questions. We are glad to hear that you find our work "well-written and well-motivated". Please kindly see below for our response.
>
> **The need for conformal prediction and calibration**
>
> Thank you for this question. We would like to clarify that the central research problem of this work is not conformal prediction or calibration, but how to interactively guide LLMs using Information Pursuit (IP). Our use of conformal prediction serves only as a mean to improving the quality of IP for LLMs.
>
> More specifically, by the definition of IP, this requires estimating the mutual information between the random variables $q(X)$ and $Y$. Our motivation for using conformal prediction is that:
> 1) computing mutual information (or conditional entropy) directly is challenging for complex distributions, especially here $q(X)$ is a distribution of text;
> 2) LLM probabilities are inaccurate (miscalibrated), which leads to poor estimates of conditional entropy and suboptimal predictive performance (as in our introduction);
> 3) Conformal Prediction provides theoretically justified and rigorous Fano-type upper bound to conditional entropy (for any $\alpha$), per our referenced proposition.
> 4) As we will discuss below, empirically we observe that C-IP performs better than IP with calibration methods.
> Hence, our work here derived the new formulation of C-IP, which leverages aspects of conformal prediction to select informative queries strategically.
>
> **Additional Baselines with Calibration Methods (Weakness and Question 3)**
>
> Thank you for the additional baseline suggestion. We agree that there is the possibility that the alternative method of "first calibrating LLM probabilities then estimate uncertainty using entropy" might find success.
> To test this hypothesis, we performed additional experiments by first applying temperature scaling (TS) and Platt scaling (PS) to calibrate LLM probabilities, then inference with standard IP (not the conformal version). Note that TS (for a fixed temperature) does not depend on calibration data, whereas PS does depend on calibration data. We evaluate 20Q in two of the open query set settings as before, and average each accuracy curve over 5 random seeds.
>
> *Open query set, Binary query answers*
> | Iteration             |        0    |        1 | 2    | 3                | 4    | 5    | 6    | 7    | 8    | 9    | 10   | 11   | 12   | 13   | 14   | 15   | 16   | 17   | 18   | 19   | 20   |
> |-----------------------|-------------|----------|------|------------------|------|------|------|------|------|------|------|------|------|------|------|------|------|------|------|------|------|
> | IP w/ TS ($T=0.5$)    | 0.05        | 0.12     | 0.2  | 0.2              | 0.26 | 0.29 | 0.29 | 0.29 | 0.31 | 0.33 | 0.34 | 0.34 | 0.36 | 0.36 | 0.39 | 0.41 | 0.41 | 0.43 | 0.43 | 0.44 | 0.45 |
> | IP w/ TS ($T=0.75$)   | 0.05        | 0.07     | 0.11 |             0.11 | 0.14 | 0.17 | 0.21 | 0.23 | 0.26 | 0.27 | 0.29 | 0.29 | 0.31 | 0.32 | 0.32 | 0.32 | 0.32 | 0.34 | 0.36 | 0.37 | 0.37 |
> | IP w/ TS ($T=1.25$)   | 0.05        | 0.1      | 0.18 | 0.26             | 0.27 | 0.3  | 0.35 | 0.38 | 0.41 | 0.42 | 0.42 | 0.42 | 0.44 | 0.46 | 0.46 | 0.47 | 0.48 | 0.48 | 0.48 | 0.49 | 0.49 |
> | IP w/ Platt Scaling   |        0.05 | 0.17     | 0.23 | 0.27             | 0.3  | 0.33 | 0.4  | 0.42 | 0.43 | 0.45 | 0.46 | 0.46 | 0.47 | 0.47 | 0.48 | 0.51 | 0.51 | 0.51 | 0.51 | 0.52 | 0.53 |
> | C-IP ($\alpha = 0.1$) |        0.05 | 0.13     | 0.19 | 0.23             | 0.31 | 0.35 | 0.41 | 0.43 | 0.44 | 0.45 | 0.46 | 0.46 | 0.48 | 0.51 | 0.52 | 0.54 | 0.57 | 0.58 | 0.58 | 0.59 | 0.59 |
>
> *Open query set, Free-text query answers*
> | Iteration              |        0    |        1 | 2    | 3    | 4    | 5    | 6    | 7    | 8    | 9    | 10   | 11   | 12   | 13   | 14   | 15   | 16   | 17   | 18   | 19   | 20   |
> |------------------------|-------------|----------|------|------|------|------|------|------|------|------|------|------|------|------|------|------|------|------|------|------|------|
> | IP w/ TS ($T=0.5$)     | 0.05        | 0.25     | 0.44 | 0.49 | 0.57 | 0.61 | 0.69 | 0.71 | 0.74 | 0.81 | 0.81 | 0.84 | 0.84 | 0.84 | 0.84 | 0.85 | 0.85 | 0.85 | 0.85 | 0.85 | 0.86 |
> | IP w/ TS ($T=0.75$)    | 0.05        | 0.28     | 0.44 | 0.56 | 0.62 | 0.62 | 0.64 | 0.69 | 0.69 | 0.71 | 0.75 | 0.77 | 0.79 | 0.79 | 0.79 | 0.79 | 0.79 | 0.79 | 0.79 | 0.79 | 0.79 |
> | IP w/ TS ($T=1.25$)    | 0.05        | 0.34     | 0.47 | 0.57 | 0.61 | 0.66 | 0.69 | 0.72 | 0.78 | 0.81 | 0.81 | 0.84 | 0.84 | 0.84 | 0.84 | 0.84 | 0.85 | 0.85 | 0.86 | 0.86 | 0.86 |
> | IP w/ Platt Scaling    |        0.05 | 0.3      | 0.43 | 0.52 | 0.61 | 0.64 | 0.74 | 0.75 | 0.77 | 0.81 | 0.82 | 0.84 | 0.84 | 0.88 | 0.89 | 0.89 | 0.9  | 0.91 | 0.91 | 0.91 | 0.91 |
> | C-IP ($\alpha = 0.15$) |        0.05 | 0.35     | 0.53 | 0.6  | 0.67 | 0.72 | 0.75 | 0.8  | 0.81 | 0.83 | 0.86 | 0.88 | 0.88 | 0.89 | 0.9  | 0.91 | 0.91 | 0.91 | 0.91 | 0.91 | 0.91 |
>
> For TS, logits obtained from LLMs are divided by a temperature scalar $T \in \\{0.5, 0.75, 1.25\\}$ before applying softmax. We observe that C-IP outperforms IP with TS in all settings.
>
> For standard calibration methods that require calibration data (e.g. PS), we note that we cannot directly "plug-in" and apply; similar to C-IP with marginal guarantee over only samples, they still require us to obtain samples with the exact history $q_{1:k}(x^{\text{obs}})$ during inference (See L190-195 for our original, full argument). Hence, to run Platt scaling with calibration data, we also need to do random sampling as for our method. In our additional experiments with 20Q, we found that IP with PS performs worse in the earlier iterations but comparably in later iterations for free-text query answers. Moreover, IP with PS performs worse in all iterations of the binary query answer setting.
>
> Overall, our additional baseline results find that C-IP outperforms IP with calibration methods of TS and PS. We argue that in the open query set settings, where the distribution is more complex, conformal prediction is a better method for good estimations of uncertainty, showing that calibration alone is not sufficient and our formulation is necessary.
>
>
> **Clarification 1: Confusion about notation and computation**
>
> Thank you for the clarification question. We agree that "some constants" need to be taken into account.
>
> The bound we showed is derived from Correira et al. 2021 Corollary 3.1 and Equation (7). The precise bound is the following. For $\alpha \in (0, 0.5)$, suppose $P_{X, Y} ( Y \in C(X)  \mid H_k ) = 1 - \alpha$ holds, then
> $$H(Y | X) \leq h_b(\alpha) + \alpha E_{X, Y \mid Y \not \in C (X)} [\log ( | {Y} | - | {C}(X) | )] + (1 - \alpha) {E}_{X, Y \mid Y \in {C}(X)} [\log | {C}(X)|]$$
> where $h_b$ is the binary entropy function. Therefore, $\alpha$ is a hyperparameter to be chosen between $(0, 0.5)$. In practice, the choice of $\alpha$ is ideally small. In our experiments, we evaluate C-IP by searching over multiple $\alpha$'s and find the one that sufficiently captures the uncertainty $Y$ at each iteration. We show that the bound is not vacuous as long as the desired coverage $\alpha$ is close to the empirical coverage (which we evaluate using the test set in Figure 3 (right column) as well as Appendix Figure 6, 7 (bottom row).
>
>
> For the second part of the question, recall that the conditional entropy $H(Y | X)$ depends only on the distribution of $X$, and not on the realized value $x$ of $X$. Hence, neither the left or right hand side depend on the values of $X$.
> However, when we apply this bound, we use it conditional on the (non-random) query $q$. So, it depends on the information provided by the query $q$ about the input $X$ (but not on the specific value of $x$), which is what we want. Then, we can optimize this over the query $q$.
>
>
> **Question 2: Implementation of Conditional Entropy and Expected Set Size (Question 2)**
>
> Using our relaxation in Section 3.2 (depending on the setting), we can find the prediction set using Algorithm 1 in Appendix I.
>
> During inference, we assume we have access to an estimation dataset $D_{est} = {\\{(x_{i}, y_{i})\\}}\_{i=1}^{N_{est}}$. Now, with $ q\_{1:k}(x^{obs}) $ in our history, we estimate $E_X [ | C_{\tau} ( q(X) ) | | H ] = \sum^{N\_{est}}_{i=1} |{C}({q\_{1:k}(x^{obs}) \cup q(x\_i)})|$. We will update and clarify this in the Appendix.
>
> Furthermore, for the second part of your question, suppose $q$ is a query that is irrelevant, say "How many wheels does the animal have?", then this query is naturally uninformative for the task, hence is unlikely to be selected. In other words, our method already takes the possibility of uninformative, irrelative queries into consideration when estimating uncertainty.

---

> > ### Comment · Reviewer_HBRz · 2025-08-07
> >
> > Thanks for the clarifications and new results --- I am raising my score as a result. For TS, why use fixed values and not optimize for LL on the calibration data though? The temperature should be able to be optimized like the platt-scaling parameters.

---

> > > ### Author Response · Authors · 2025-08-07
> > >
> > > Thank you for the follow-up question. We agree that temperature scaling we showed in our rebuttal can be optimized with calibration data as well, but we wanted to show consistency across different settings by searching through the same set of temperatures. Hence, we had the same temperature choices $\tau$'s for all of our settings. In our updated version, we will show the optimal temperature based on the calibration data for every setting.

---

> ### Author Response · Authors · 2025-08-04
>
> Thank you again for the insightful reviews. If there are any unresolved concerns and questions, feel free to ask any additional questions to which we may answer and provide additional clarification. If you feel our rebuttal has sufficiently addressed your concerns, we would appreciate it if you could kindly reconsider your evaluation.
>
>
>
> Sincerely,
>
> Authors

---

### Note · Authors · 2025-08-11

We thank the area chair and the reviewers for offering their feedback on our paper. We are glad to hear that reviewers find our paper: 1) clearly motivated, theoretically sound and rigorous foundation (QsyY, zf88); 2) demonstrated practical effectiveness (HBRz, zf88); 3)well-written and easy-to-follow (HBRz, spQy, QsyY). We thank the reviewers for acknowledging the significance and novelty of our work.

In this work, we proposed a novel method of Conformal Information Pursuit (C-IP), where reduces the number of LLM interactions via the well-accepted principle of information gain. One of our core contributions is that our derivation of the C-IP is also theoretically-grounded with connections to proven theorems. Relative to existing literature, our work provides a potential answer to efficient interactions with LLMs from an information-theoretic perspective, which, to the best of our knowledge, has not been well-explored. The significance of our work is highlighted, with Reviewer QsyY claiming that "reducing the number of questions in such interactive LLM-settings is an important research aim" and Reviewer zf88 claiming that our work "makes a significant contribution to uncertainty quantification in sequential querying through rigorous mathematical foundations." Overall, we argue that our work provides the first of many steps towards a principled and broadly applicable framework towards interactive ML and uncertainty quantification.

During the rebuttal, one core concern was whether simpler calibration methods would suffice over our proposed method (HBRz, spQy). From our additional results, we found that C-IP outperforms our existing calibration methods (i.e. Temperature Scaling and Platt Scaling), demonstrating the need for a better method of selecting queries. We believe we have sufficiently addressed the reviewers' concern, as Reviewer HBRz has also acknowledged our updated results and have raised the score consequently.

We would also like to again address the concerns that Reviewer spQy has brought up. Namely, 1) our work is not similar to Conformal Language Modeling; 2) we have compared to proper baselines, including IP, UoT, and provided additional baselines for calibration; and 3) the core contribution of our work is to propose an algorithm for interactive LLMs, hence the fully formal theoretical results can be reserved for future work. Unfortunately, Reviewer spQy did not respond to our rebuttal or provide any justification for the final rating.

---

### Decision · Program_Chairs · 2025-09-17

**Decision:**

Accept (poster)

**Comment:**

The paper proposes an approach for efficient multi-step problem solving for LLMs. The approach aims to ask the most “informative” query at each iteration given prior history, until the model is confident enough to make a prediction.

They use Information Pursuit framework and define the most informative query as the one that maximizes information gain at each iteration, or equivalently the query whose answer minimizes the uncertainty of the prediction. They remark that estimating conditional entropy for this purpose is challenging due to the fact that the LLM outputs are often miscalibrated and propose an alternative approach to quantify sequential information gain that leverages the notion of prediction sets from conformal prediction. They argue that prediction sets from conformal prediction leads to more robust estimates compared to standard entropy. They utilize conformal prediction sets based on probability estimates from the LLM, and at each iteration seek to minimize the prediction set size as LLM gains more information at each turn.

The problem is clearly motivated and the approach is theoretically founded. The paper demonstrates practical utility of the proposed approach via experiments showing reduced number of questions required by the proposed approach. Discussions during the rebuttal period helped address some of the reviewer concerns (e.g. how simpler calibration methods would compare to proposed approach, clarification about notation, clarification on bias of prediction sets), with general agreement amongst reviewers on the technical soundness and supporting empirical results.